# Is Positive Correlation between Cloud Droplet Effective Radius and Aerosol Optical Depth over Land Due to Retrieval Artifacts or Real Physical Processes?

Hailing Jia[1, 2], Xiaoyan Ma[1, 2], Johannes Quaas[3], Yan Yin[1, 2], and Tom Qiu[4]

[1]Collaborative Innovation Center on Forecast and Evaluation of Meteorological Disasters, Nanjing University of Information Science & Technology, Nanjing 210044, China
[2]Key Laboratory for Aerosol-Cloud-Precipitation of China Meteorological Administration, School of Atmospheric Physics, Nanjing University of Information Science & Technology, Nanjing 210044, China
[3]Institute for Meteorology, Universität Leipzig, Leipzig, Germany
[4]University of Victoria, Victoria, Canada

*Correspondence to*: Xiaoyan Ma (xma@nuist.edu.cn)

**Abstract.** The Moderate Resolution Imaging Spectroradiometer (MODIS) C6 L3, Clouds and the Earth's Radiant Energy System (CERES) Edition-4 L3 products, and the European Centre for Medium-Range Weather Forecasts (ECMWF) ERA-Interim reanalysis data are employed to systematically study aerosol-cloud correlations over three anthropogenic aerosol regions and their adjacent oceans, as well as explore the effect of retrieval artifacts and underlying physical mechanisms. This study is confined to warm phase and single layer clouds without precipitation during the summertime (June, July, and August). Our analysis suggest that cloud effective radius (CER) is positively correlated with aerosol optical depth (AOD) over land (positive slopes), but negatively correlated with aerosol index (AI) over oceans (negative slopes) even with small ranges of liquid water path (quasi-constant). The changes in albedo at the top of atmosphere (TOA) corresponding to aerosol-induced changes in CER also lends credence to the authenticity of this opposite aerosol-cloud correlation between land and ocean. It is noted that potential artifacts, such as the retrieval biases of both cloud (partially cloudy and 3-D shaped clouds) and aerosol, can result in a serious overestimation of the slope of CER-AOD/AI. Our results show that collision-coalescence seems not to be the dominant cause for positive slope over land, but the increased CER caused by increased aerosol might further increase CER by initializing collision-coalescence, generating a positive feedback. By stratifying data according to the lower tropospheric stability and relative humidity near cloud top, it is found that the positive correlations more likely occur in case of drier cloud top and stronger turbulence in clouds, while negative correlations occur in case of moister cloud top and weaker turbulence in clouds, which implies entrainment mixing might be a possible physical interpretation for such a positive CER-AOD slope.

# 1 Introduction

Atmospheric particles influence the Earth's radiation budget and hence climate change by directly scattering and absorbing solar radiation and indirectly by acting as cloud condensation nuclei (CCN), altering cloud properties and precipitation (Twomey, 1977; Albrecht, 1989). The latter historically has been referred to as the aerosol indirect effect, and more recently as the effective radiative forcing due to aerosol-cloud interactions, which remain to have the largest uncertainty in assessing the anthropogenic contribution to present climate change (IPCC, 2013). An increase in CCN number concentration will

generate a cloud that consists of more but smaller drops–under constant cloud liquid water path (LWP). The consequence is scattering of more solar radiation back to space. The decrease in cloud effective radius (CER) with increasing aerosol concentrations was historically termed aerosol first indirect effect (AIE), cloud albedo effect, or Twomey effect (Twomey, 1977: Feingold et al., 2003), and in the more recent literature, as radiative forcing due to aerosol-cloud interactions (IPCC, 2013).

There have been many observational evidences from in-situ aircraft measurements (Pawlowska & Brenguier, 2000; Wilcox et al., 2006; Roberts et al., 2008; Kleinman et al., 2012; Werner et al., 2014), ground- (Feingold et al., 2003; Kim et al, 2003; Garrett et al., 2004; Qiu et al., 2017) and satellite-based (Nakajima et al., 2001; Bréon et al., 2002; Kaufman et al., 2005; Koren et al., 2005, 2010; Quaas et al., 2008; Costantino & Bréon, 2010, 2013; Lihavainen et al., 2010; Chen et al., 2014; Christensen et al., 2016) remote sensing in support of this negative correlation between aerosol concentrations and CER. Moreover, solid

evidences for Twomey effect were also found from aerosol-induced changes in top of atmosphere (TOA) albedo. Su et al. (2010) found a significant increase in TOA albedo associated with continental aerosols relative to those associated with oceanic aerosols under all LWP ranges when cloud fraction is constrained; similar results have been reported by Chen et al. (2014) and Christensen et al. (2016). However, relationships of aerosol optical depth and aerosol index with cloud albedo, cloud fraction, and cloud liquid water path have been shown to be difficult to interpret due to the confounding influence of relative humidity

fluctuations that impact both quantities (Quaas et al., 2009, 2010; Gryspeerdt et al. 2014; 2016).

In addition to the explorations on aerosol-cloud interactions at the cloud top (from satellite-based remote sensing instruments) and cloud base (from ground-based remote sensing instruments), the role of vertical observations in the cloud-aerosol-precipitation interaction has been studied by using satellite-based radar/lidar. For example, the measurements provided by CALIOP sensors were widely used to obtain the respective position of aerosol and cloud layers (Costantino and Bréon,

2010, 2013; Zuidema, et al., 2016; Liu et al., 2017), which can improve the estimation of the amount of aerosols that actually enter the cloud. The CPR (cloud profiling radar) onboard CloudSat is able to penetrate optically thick clouds layers (Wang et al., 2013), and thus has been used to differentiate cloud regimes (Peng et al., 2015; Christensen et al., 2016) and investigate the response of cloud vertical structure to aerosols (Chen et al., 2016). Additionally, the aerosol-induced changes in the vertical structure of precipitation were also explored by using the Tropical Rainfall Measuring Mission (TRMM) products (Wall et al.,

2014; Li et al., 2016; Guo et al., 2018).

In addition to the widely observed Twomey effect, positive correlations between aerosol concentrations and CER were also found in some regions, such as southeastern US and southeastern China (Yuan et al., 2008), eastern China (Tang et al., 2014; Wang et al., 2014, 2015; Liu et al., 2017), and Indian (Panicker et al., 2010; Manoj et al., 2012). Overall, positive correlations occur over land while negative correlations dominate over ocean (Grandey and Stier, 2010; Ma et al., 2018). The lack of consensus on these relationships motivates further exploration of underlying physical reasons for these opposite correlations.

It is acknowledged that quantification of aerosol first indirect effect is highly uncertain due to various influencing factors, including (1) aerosol microphysics such as size (Feingold et al., 2001; Dusek et al., 2006; Zhang et al., 2011), chemical composition (Nenes et al., 2002; Lance et al., 2004; Ervens et al., 2005; McFiggans et al., 2006; Almeida et al., 2014) and mixing state (Wang et al., 2008, 2010), (2) meteorological conditions such as vertical velocity (Koren et al., 2010; Lu et al., 2012), lower tropospheric stability (Wang et al., 2014; Saponaro et al., 2017; Ma et al., 2018), wind shear (Fan et al., 2009), and precipitable water vapor (Yuan et al., 2008; Qiu et al., 2017), (3) cloud types (Gryspeerdt and Stier, 2012; Chen et al., 2016), and (4) vertical overlapping status of aerosol and cloud layers (Costantino and Bréon, 2010, 2013; Huang et al., 2015; Liu et al., 2017). It is extremely difficult to completely isolate the response of CER to aerosol perturbations from the above-mentioned influencing factors. Yuan et al. (2008) examined the positive correlation between CER and aerosol optical depth (AOD) by using Moderate Resolution Imaging Spectroradiometer (MODIS) satellite products, and speculated that slightly soluble organics particles, which induced a decrease of aerosol activation, might be a possible explanation for the positive correlation. Wang et al. (2014) explored AIE over Eastern China by using both MODIS satellite and National Center for Environmental Prediction (NCEP) reanalysis data and found that positive correlations are more likely to be found under unstable atmosphere conditions. Ma et al. (2018) employed MODIS satellite products and ERA-Interim reanalysis data to systematically explore the impact of meteorological conditions on the correlations over the major industrial regions and over relatively clean oceans, and concluded that positive correlations are more likely corresponding to relatively high cloud top height (CTH) and low lower tropospheric stability (LTS), while negative relationships were predominantly found for low CTH and high LTS. Tang et al. (2014) pointed out that covariation of wind field and relative humidity in North China Plain may contribute to such positive correlations, that is, relatively wet and polluted southerly wind leads to simultaneous increases in both AOD and CER, while dry and clean northerly wind results in coincident decreases in AOD and CER. Gryspeerdt and Stier (2012) reported that the strongest positive correlation occurs in the shallow cumulus cloud regimes, while the negative one is analyzed for the stratiform cloud regimes. By using the profiles from the CALIPSO lidar that identifies the respective position of aerosol and cloud layers, Costantino and Bréon (2010, 2013) found that the aerosol indirect effect is stronger for well-mixed aerosol and cloud layers than for separated one.

In addition to real physical-chemical processes stated above, the positive correlations between AOD (or Aerosol Index; AI) and CER may also result from artificial correlations due to the biases of the retrievals. The MODIS aerosol retrieval algorithm is conducted only for clear pixels determined by a cloud mask. AOD could be overestimated due to either cloud contaminations where spurious clouds might be present in pixels that are erroneously identified as completely clear pixels (Kaufman et al.,

2005; Remer et al., 2005; Zhang et al., 2005), or cloud adjacency (or 3-D) effect where cloud-free pixels are brightened by reflected light from surrounding clouds (Cahalan et al., 2001; Wen et al., 2006, 2007; Várnai and Marshak, 2009). Moreover, cloud retrievals applied to partially cloudy and 3-D shaped clouds pixels are expected to deviate from the retrieval assumptions of overcast homogenous cloud and 1D plane-parallel radiative transfer, and tend to result in overestimation of CER (Han et al., 1994; Coakley et al., 2005; Matheson et al., 2006; Zhang and Platnick, 2011; Zhang et al., 2012; Grosvenor et al., 2018).

Therefore, covariation of biases in CER and AOD (AI) may incur a false correlation between the two variables.

As stated above, aerosol-cloud correlations derived from satellite remote sensing are potentially veiled by large variations in both retrieval biases and meteorological conditions. By employing MODIS and CERES satellite data as well as ERA-Interim reanalysis data, this study aims to: (a) examine whether the positive relationship between AI and CER over land is true, (b) assess how and to what extent the satellite retrieval biases may affect the satellite-diagnosed aerosol-cloud correlations, and

(c) explore the underlying physical mechanisms. This paper is organized as follows: the data descriptions of both satellite and reanalysis and data processing are presented in Sect. 2, and major findings are discussed in Sect. 3. A summary and discussion is given in Sect. 4.

## 2 Data

In this study, three regions with strong anthropogenic emissions over land, namely, East China (EC), East U.S. (EU), and

West Europe (WE), as well as their neighboring oceans (ECO, EUO, and WEO, respectively), are chosen to systematically examine aerosol-cloud correlations under different anthropogenic emissions and dynamic and thermodynamic conditions (Fig. 1). The data used include aerosol and cloud properties gathered from the MODIS/Aqua Collection 6 Level-3 (L3) daily product (Levy et al., 2013; Platnick et al., 2017), albedo at the top of the atmosphere (TOA) obtained from the Clouds and the Earth's Radiant Energy System (CERES) Aqua Edition-4 L3 products (Wielicki et al., 1996), as well as meteorological variables

extracted from the European Centre for Medium-Range Weather Forecasts (ECMWF) ERA-Interim reanalysis data (Dee et al., 2011). The 14 consecutive years (2003–2016) of daily $1° \times 1°$ gridded data are used for statistical analysis in order to obtain statistically significant results from enough samples.

## 2.1 Satellite Data

The MODIS C6 L3 product provides AOD retrieved at several wavelengths globally (Levy et al., 2013; Sayer et al., 2014),

which has been validated extensively (Remer et al., 2005; Tripathi et al., 2005). In this study, the AOD products retrieved from the Dark Target (DT) algorithm are used. Aerosol index (AI), in comparison to AOD, is believed to be a better proxy for CCN since it contains the information of aerosol size (Nakajima et al., 2001; Stier, 2016). The AI can be derived on the basis of AOD and Ångström exponent (AE), with the former provided directly by the MODIS product and the latter calculated from AOD at wavelength of 460 and 660 nm. However, the MODIS retrieval of AE over land may be problematic (Levy et al.,

2013; Sayer et al., 2013). For collection 6 products, several previous studies have found little quantitative skill in MODIS-

retrieved aerosol size parameters over land (Levy et al., 2010; Mielonen et al., 2011). For this reason, we use AOD and AI as proxies for CCN over land and ocean, respectively.

The MODIS C6 L3 product provides cloud macrophysical parameters, including cloud fraction (CF), cloud top temperature, cloud top pressure, and cloud top height, as well as cloud microphysical parameters (CER, LWP, and cloud optical depth) with
statistics (mean, minimum, maximum, and standard deviation) at three wavelengths (1.6, 2.1, and 3.7 μm) for individual cloud phases (liquid, ice, and undetermined) separately. We filtered the MODIS cloud data according to the criteria employed by Saponaro et al. (2017) to ensure the data used for analysis are only limited to warm liquid phase and single layer clouds, and non-precipitating cases. In previous versions, before MODIS C6, all pixels identified as partly cloudy (either partially cloud-covered or at cloud edge) were restored to clear sky, and the corresponding cloud retrievals were missing, that is, cloud
retrievals were only performed in overcast cloudy pixels. In C6, however, the retrievals of cloud microphysical properties are now attempted on these pixels, and successful retrievals are reported in the Level-2 product and aggregated to the Level-3 product, which are reported separately in partly cloudy (PCL) science data sets (SDSs) and are segregated from the normal "overcast" SDSs. Therefore, the simultaneous availability of PCL and normal "overcast" SDSs provides a great opportunity to investigate the effect of partly cloudy retrievals on aerosol-cloud correlations.

The CERES SSF1deg Edition-4 product provides radiative fluxes and albedo at TOA for all-sky and clear conditions in the longwave, shortwave, and window regions (Loeb et al. 2005). The albedo at TOA (plenary albedo) is the ratio of broadband (0.2-5μm) shortwave reflected and the incoming solar flux at the top of the atmosphere. In comparison with the previous version, the CERES instrument calibration and CERES instrument spectral response function corrections have been improved in Edition-4 (Doelling et al., 2016a, 2016b).

The satellite-based measurements can provide long-term data sets of aerosols and clouds in high spatial and temporal resolution on a global scale, and have been believed to be a promising tool for the study of aerosol-cloud interactions. However, several inherent limitations prevent a definite quantitative interpretation of aerosol indirect effects. The relationship between aerosol and cloud droplet number, $N_d$, at cloud base fundamentally is controlled by both the critical supersaturation of aerosol particles, which described by Köhler theory, and the maximum supersaturation at cloud base, which determined by the updraft
speed and the competition of the existing CCN for the available humidity. $N_d$ at cloud base thus is a function of cloud-scale, cloud-base vertical wind, aerosol size distribution, and solubility. $N_d$ at cloud base thus necessarily is positively correlated to CCN, with an approximately logarithmic relationship. In this study, we use the satellite-retrieved cloud-top (at approximately 1~5 optical depth into the cloud, depending on the wavelength employed to retrieve it (Platnick, 2000)) CER stratified by classes of LWP as cloud quantity. It deviates from cloud-base $N_d$ since
- the stratification by LWP bins might not be sufficient to disentangle LWP and $N_d$ impacts on CER

    - cloud-top $N_d$ might be related only loosely to cloud-base $N_d$ in case the clouds are non-adiabatic (entrain environmental air from the sides or top), and/or coagulation of cloud droplets occurs

    - the retrieved CER might be contaminated by various retrieval problems and thus is possibly only loosely related to the real cloud-top CER.

As aerosol quantity, we use AOD/AI retrieved in the pixels determined as cloud-free in the retrieval algorithm. This may be only loosely related to the cloud-base CCN. Stier (2016) stated that 52 % of the area of the globe shows correlation coefficients between $CCN_{0.2\%}$ at cloud base and AI below 0.5. This is largely due to:

- AOD/AI is a vertical integral and so might be dominated by aerosol layers that do not affect cloud-base CCN concentrations (Stier, 2016)

- the clear-sky column retrievals might not be representative of the cloud-base CCN in the neighbouring cloudy pixels

- AOD/AI might be loosely related to cloud-base CCN if affected by deliquescence in high relative humidity environments, or if dust or other insoluble aerosol has a substantial share of the total aerosol.

## 2.2 Reanalysis Data

To explore the extent to which meteorological conditions affect the CER correlation with AOD/AI, ERA-Interim reanalysis data was employed to derive the lower tropospheric stability (LTS) and relative humidity near cloud top ($RH_{CT}$). LTS is computed as the difference between the potential temperature at 700 hPa and at the surface (Klein and Hartmann, 1993), representing the magnitude of the inversion strength in the lower troposphere. Large LTS is associated with stable conditions in which vertical mixing is suppressed . $RH_{CT}$ is the relative humidity over the pressure levels closest to the cloud top pressure. The daily $1° \times 1°$ gridded reanalysis data at 14:00 local solar time are used to match the cloud parameters obtained from MODIS L3 data.

## 3 Results

## 3.1 Correlations between CER and AOD/AI

According to Twomey (1977), the presence of anthropogenic aerosols increases the cloud droplet number concentration but decreases CER for a constant LWP. Since CER is a function of both AOD/AI and LWP, under conditions where LWP changes with aerosol concentration, the variation in LWP could act to modulate the relationship between AOD/AI and CER. Therefore, the constant LWP assumption should be highlighted in assessing aerosol first indirect effect. However, many previous studies did not constrain the LWP (Breon et al., 2002; Tang et al., 2014; Wang et al., 2014, 2015; Liu et al., 2017), or constrained the LWP into coarse intervals (Nakajima et al., 2001; Sekiguchi et al., 2003; Yuan et al., 2008), which would induce the uncertainty in correlating CER and AOD/AI. In our previous study (Ma et al., 2018), the CER and AOD/AI are grouped over LWP bins with an interval of 40 g m$^{-2}$, and then a linear regression analysis with the logarithms of the CER and AOD/AI in each LWP bin is performed (as shown in Fig. 2a). It is found that CER is positively correlated with AOD over land (positive slopes), but negatively correlated with AI over their adjacent oceans (negative slopes), and the positive slopes over land become weaker while negative slopes over ocean change slightly as LWP increases. To examine whether such intervals of LWP bin can result in the uncertainty in quantifying CER-AOD/AI correlation, especially the positive ones over land shown in Fig. 2a, we conducted a similar statistical analysis for the LWP versus AOD/AI within each LWP bin. As shown in Fig. 2b,

in the smallest LWP bin (20-60 g m$^{-2}$, thin cloud), LWP is more sensitive to AOD/AI, with a significant positive correlation over land but negative correlation over ocean, which will amplify the positive CER-AOD correlation over land and the negative CER-AI correlation over ocean shown in Fig. 2a. For a larger LWP bin (>100 g m$^{-2}$), LWP is insensitive to both AOD and AI, indicating that the increase of cloud water is governed by meteorological conditions instead of aerosol in case of thick clouds.
Therefore, coarse intervals of LWP bin might be a possible cause for the appearance that the positive slopes of CER versus AI over land become weaker as LWP increases (Ma et al., 2018).

It is noted that the positive CER-AOD correlation over land can still be found even within bins where LWP is insensitive to AI, implying that the variation of LWP within bin may not the dominant cause for the observed positive relationship over land for large LWP. However, for the smaller LWP bin where LWP is susceptible to AI, it is still not clear whether the sign of
CER-AI correlation is interfered by LWP changes. To this end, we constrain the LWP to a smaller range (5 g m$^{-2}$) to ensure more constant LWP. As data samples in such small intervals are insufficient to calculate the slope of CER versus AOD/AI, the difference of CER between polluted and clean conditions was thus employed as an alternative to the slopes. In this study, two-dimensional ("joint") histograms of CER and LWP are used to show the probability distribution function of CER at each LWP bin. Meanwhile, the 25th and 75th percentiles of the AOD/AI are computed for each bin with certain CER and LWP,
and then the samples with AOD/AI lower than 25th percentiles and with AOD/AI greater than 75th percentiles are classified as 'clean case' and 'polluted case', respectively. The difference between the joint histograms in polluted and clean cases demonstrates how CER varies with AOD over land and AI over ocean in a quasi-constant LWP bin. Figure 3 indicates that CER increases significantly with increasing LWP over all regions, and the dependence of CER on LWP in polluted case is weaker than that in clean case. By looking at the difference between the joint histograms in polluted and clean cases (the third
column in Fig. 3), it is clearly shown that, over land, the samples of polluted case tend to concentrate in the larger CER bin than clean case, while the opposite is found over ocean. That is, as AOD/AI increases, CER increases over land but decreases over ocean even with the tighter constraint on LWP. It can thus be concluded that the variation of LWP within a bin do not change the sign of CER-AOD/AI slope, although it would overestimate the correlation between CER and AOD/AI to some extent.

Additionally, previous studies have noted a nonlinear effect of aerosols on cloud microphysics (Breon et al., 2002; Lohmann and Lesins, 2002; Jin et al., 2008; Gryspeerdt et al., 2017). Using the polarization and directionality of the earth reflectances (POLDER) satellite data, Breon et al. (2002) found that the slope of CER versus AI on a global scale is highly significant for AI less than 0.15, whereas a saturation effect occurs as the aerosol keeps rising. The similar saturation effect was also reproduced by the simulations in global climate models (Lohmann and Lesins, 2002; Gryspeerdt et al., 2017). In this
study, such saturation effect was found in three ocean regions (ECO, EUO, and WEO) adjacent to the anthropogenic regions (Fig. S1), but with a higher threshold (AI = 0.3) than that derived from the analysis on global oceans (Breon et al., 2002). However, it is noted that no saturation effect was found in the regions with strong anthropogenic emissions over land (EC, EU, and WE), i.e., CER shows a continuous increase as AOD increases (Fig. S1), suggesting that the saturation effects does not appear to interfere with the observed positive slopes over land.

## 3.2 Evidences for the Positive Correlation between CER and AOD over Land

Although the positive correlation between CER and AOD/AI over land has been reported by several satellite-based studies (Tang et al., 2014; Wang et al., 2014, 2015; Liu et al., 2017; Ma et al., 2018;), its reliability is still controversial due to the limitations of satellite retrievals. The main concerns include: (a) satellite-retrieved aerosol and cloud properties remain to have biases; (b) the cloud properties observed from satellites are confined to cloud tops rather than entire cloud; and (c) the AI or AOD may not actually represent the magnitude of aerosols entering the clouds. Therefore, it is necessary to confirm whether the positive correlation is real before discussing possible physical causes.

Under cloudy sky, the response of TOA in-cloud albedo at a given LWP on aerosol changes is considered to be a more direct indicator for the Twomey effect than CER (Su et al., 2010), given that the albedo is calculated from CERES TOA upwelling fluxes while CER is retrieved from MODIS reflectance based on multiple assumptions and thus tends to have larger retrieval biases (Matheson et al., 2006; Vant-Hull et al., 2007; Zhang and Platnick, 2011; Zhang et al., 2012). In addition, aerosol and cloud retrievals can also interfere with each other, and thus have potential to result in an artifact correlation. These retrieval biases will be discussed in more detail in section 3.3.1. Therefore, using TOA albedo ($\alpha$) as a proxy to examine Twomey effect can better exclude the influence of artifacts in the retrievals. In general, the variations of all-sky TOA albedo are controlled by the following five parts: (a) cloud fraction (CF, representing the horizontal extent of cloud); (b) cloud liquid water path (representing the vertical development of a cloud); (c) cloud droplet size (that is, the Twomey effect when considering aerosol perturbations at a certain LWP); (d) clear-sky aerosol loading and above-cloud absorbing aerosol, and (e) surface albedo. In the presence of clouds, the aerosol-induced TOA albedo changes are minimal compared to cloud-induced changes, and therefore (d) are not considered in our analysis. Additionally, for a specific season and region, surface albedo is also assumed to be uniform (Robock, 1980; Wang et al.,2004), so the effect of (e) is ruled out. In this study, CERES TOA albedo and AI are stratified according to CF (with an interval of 0.04) and LWP (with an interval of 10 g m$^{-2}$) in order to isolate the Twomey effect from the interference by them, and 'clean case' and 'polluted case' are distinguished according to the method used in section 3.1. Thereby, for a specific bin determined by both CF and LWP, the difference of TOA albedo between polluted and clean conditions ($\Delta\alpha$) can be used to represent the changes in TOA albedo caused only by the Twomey effect (Fig. 4). The number of samples for each LWP and CF bin is shown in Fig. S2. A positive $\Delta\alpha$ means that the observed TOA albedo has a larger value in the polluted case than in clean one, and the reverse is true when $\Delta\alpha$ has a negative value. Figure 4 shows that $\Delta\alpha$ is normally negative over land (top), but positive over ocean (bottom). The ratio of positive to total samples over EC , EU, and WE is only 37 %, 11 %, and 13 %, respectively, which is evidently less than that over ECO (85 %), EUO (61 %), and WEO (84 %). This result implies that the reflected solar shortwave radiation at TOA generally tends to reduce with AOD over land while to increase with AI over ocean. As mentioned above, under the constant CF and LWP, the response of TOA albedo to elevated aerosol loading can be interpreted as the result of changes in CER. That is, under polluted conditions, larger droplets over land scatter less solar radiation back to space, inducing lower TOA albedo, whereas smaller droplets over

ocean do the opposite. This finding hence lends credibility to the positive CER-AOD correlations over land and negative CER-AI correlations over ocean, as derived in section 3.1.

## 3.3 Factors Influencing the Slope of CER versus AOD/AI

### 3.3.1 Artifact Correlations

Although the positive correlations between CER and AOD over land is believed to be a real relationship as discussed in section 3.2, the magnitude of this slope may still be subject to artificial correlations due to the biases of both aerosol retrievals (cloud contamination and cloud adjacency effect) and cloud retrievals. In other words, if biases in CER are correlated with biases in AOD, a false correlation may occur between the two variables.

The MODIS aerosol retrieval algorithm is applied only for clear pixels determined by cloud mask (Remer et al., 2005). To screen out cloudy pixels, a $3 \times 3$ standard deviation test was used to detect low clouds (Martins et al., 2002) and IR channels to identify high clouds in addition to the standard cloud mask product. The 20% darkest and 50% brightest pixels were also removed for possible cloud contaminations. However, a part of pixels associated with cloud contaminations might still be erroneously identified as clear (Kaufman et al., 2005; Remer et al., 2005; Zhang et al., 2005), inducing an overestimation of

AOD. Meanwhile, cloud-free pixels may be brightened by reflected light from surrounding clouds, i.e., cloud adjacency effect (Cahalan et al., 2001; Wen et al., 2006, 2007; Varnai and Marshak, 2009), which could also overestimate AOD. The overestimation of AOD caused by both cloud contaminations and cloud adjacency effect was found to increase as cloud fraction (CF) increases (Zhang et al., 2005). Additionally, cloud retrievals applied to the partly cloudy (PCL) pixels are expected to deviate from the retrieval assumptions of overcast homogenous cloud, and tend to overestimate CER (Han et al.,

1994; Coakley et al., 2005; Matheson et al., 2006). Therefore, a spurious positive correlation could occur in case of co-existence of overestimation of AOD and CER. To look into the contribution of retrieval biases on artifact correlations, the correlations of CER and AOD/AI derived from PCL retrievals and normal "overcast" retrievals are compared, where data are stratified according to CF in order to account for the impact of biases in aerosol retrievals (Fig. 5). Considering the limited data samples for the LWP bins of 140-180 and 180-220 g m$^{-2}$, the associated statistical results are not included into our analysis. As seen in

Fig. 5, the CER-AOD/AI slopes in PCL retrieval (circles) are significantly larger than that in overcast retrieval (dots). It is interesting to see that, over ocean, where the Twomey effect dominates actually, the slope even becomes positive in the PCL case, which suggests that CER biases in PCL retrievals can result in a serious overestimation of the positive slope. In addition to PCL retrievals, the biases in aerosol retrievals could also affect this positive slope. The differences between the slope in PCL and in overcast retrievals in case of CF > 60% ($\Delta S_{|>60}$) and CF < 60% ($\Delta S_{|<60}$) over three anthropogenic regions and their

adjacent oceans are summarized in Table 1, which clearly shows that $\Delta S_{|>60}$ is overall larger than $\Delta S_{|<60}$ under all LWP bins, implying that the overestimation of AOD due to retrieval biases under larger CF can amplify the positive slope caused by PCL retrievals. Therefore, in the actual retrieval process, if MODIS algorithm erroneously identifies PCL pixels as overcast pixels, the artifact positive correlation will be introduced, especially for condition of larger CF.

In addition to PCL retrievals, the cloud retrievals for 3-D shaped clouds are also problematic due to deviating from the retrieval assumptions of plane-parallel clouds (Nakajima and King, 1990). The variations in the sun–satellite scattering geometries can interfere satellite-measured signals and hence induce biases in retrievals for 3-D shaped clouds, which are interpreted as shadowing and illumination effects (Vant-Hull et al., 2007). Such retrieval biases in CER were found to be larger in 2.1μm channel than in 3.7μm channel (Zhang and Platnick, 2011; Zhang et al., 2012). Our previous study (Ma et al., 2018) stated that the results of positive slope over land and negative slope over ocean changed little when using CER in 3.7 μm ($CER_{3.7μm}$) rather than CER in 2.1 μm ($CER_{2.1μm}$) for the regression analysis. In order to examine the effect of biases in 3-D shaped cloud retrievals on artifact correlations, the differences between CER2.1μm-AOD slope and CER3.7μm-AOD slope over land and the differences between $CER_{2.1μm}$-AI slope and $CER_{3.7μm}$-AI slope over ocean under all LWP bins are calculated, respectively (Fig. 6). The number of samples for each bin for regression analysis is shown in Fig. S3. It is clearly shown that the differences are normally positive values over both land and ocean, with the maximum difference of 0.08, which suggest that retrieval biases for 3-D shaped cloud tend to result in a higher slope than its physically correct value.

### 3.3.2 Possible Physical Explanations

### 3.3.2.1 Collision and Coalescence

In warm clouds, collision-coalescence is a much more effective process to increase droplet size compared with diffusion of water vapor (Langmuir, 1948; Kogan, 1993; Beard and Ochs, 1993; Pruppacher and Klett, 1997). The impact of collision-coalescence on CER is believed to be minimal when CER is smaller than ~14 μm, while the coalescence rate increases very rapidly when CER is greater than this value (Gerber, 1996; Freud and Rosenfeld, 2012), i.e., collision-coalescence becomes dominant. To explore the possible contribution of collision-coalescence processes to the correlation between CER and AOD/AI, the data are thus stratified into two subsets with CER greater and less than 14 μm, respectively. As shown in Fig. 7, the dependence of CER on AOD/AI becomes fairly weak with the slope close to zero over both land and ocean when collision-coalescence process is dominant (blue; CER > 14 μm), while CER-AOD/AI slopes are still positive over land and negative over ocean when collision-coalescence is negligible, which suggests that collision-coalescence is not likely the major cause for the opposite CER-AOD/AI correlations over land and ocean. Figure 8 shows the probability density functions of AOD/AI for the above two subsets over land and ocean, respectively. It is clearly demonstrated that the occurrence frequency of AOD/AI presents a significant difference between the case of CER > 14 μm and CER < 14 μm, with lager AOD for the former and smaller AOD for the later over land, while the reverse is true over ocean. This result implies that collision-coalescence are more likely to occur in case of larger AOD over land while smaller AI over ocean, which may further enhance the positive CER-AOD slope over the land and the negative CER-AI slope over ocean, as demonstrated in Fig.7. From the above analyses, we assume that although collision-coalescence is not the dominant cause for positive slope over land, the increased CER caused by increased aerosol might further increase CER by initializing collision-coalescence, generating a positive feedback.

## 3.3.2.2 Entrainment mixing

In addition to the aforementioned collision-coalescence process, entrainment mixing is also believed to be an effective way to reduce cloud droplet number concentration and increase CER, which of course depends on the specific details of entrainment-mixing mechanisms, as discussed in depth by Kim et al. (2006). The possible impacts of the entrainment mixing process on the estimated aerosol first indirect effect (AIE) have been reported by previous studies. For instance, Kim et al. (2006) found that the AIE is relatively weak in sub-adiabatic clouds compared with adiabatic clouds based on ground-based remote sensing at the Southern Great Plains, and pointed out that this may be due to interference from heterogeneous entrainment mixing that change the droplet number concentrations in a manner that attenuates the AIE. Additionally, Shao and Liu (2006) suggested the AIE is about half of those estimated by many previous studies, and attributed this difference to evaporation resulting from entrainment mixing processes. By using a mixed-layer model, Dal Gesso et al. (2014) found that the entrainment rate in stratocumulus clouds tends to be larger for the smaller lower tropospheric stability (LTS). Although LTS is an indicator of temperature inversion proposed as relevant for marine stratocumulus clouds (Klein and Hartmann, 1993), it has also been widely used to characterize atmospheric stability over both land and ocean, and thereby indicating the vertical mixing (Chen et al., 2014; Chen et al., 2016; Liu et al., 2017; Ma et al., 2018). As known, the degree of entrainment mixing is subject to not only the intensity of turbulence in clouds but also the relative humidity of air outside clouds (De Rooy et al., 2013). In this study, the former is represented by lower tropospheric stability (LTS), and the latter is described by relative humidity near cloud top ($RH_{CT}$).

To investigate the extent to which entrainment mixing affect the slopes of CER versus AOD/AI, the dataset is thus stratified according to LTS and $RH_{CT}$ to differentiate environmental regimes. In order to satisfy the constant LWP assumption for aerosol first indirect effect, LWP bin range from 60 to 100 g m$^{-2}$ is employed, among which the LWP does not change significantly with AOD/AI (as demonstrated in Fig. 1b) and statistical samples are sufficient for regression analysis. The slopes of CER versus AOD over land and the slopes of CER versus AI over ocean on log-log scale under low, medium, and high LTS and $RH_{CT}$ conditions are shown in Fig. 9, in which nine categories are classified to make sure that each category has roughly the same number of samples (Table 2). It is clearly shown that slopes under low LTS (weak stable condition) are more positive than that under medium (medium stable) and high LTS (stable condition) over both land and ocean. Also, the slopes under low $RH_{CT}$ (dry condition) are overall more positive than that under high $RH_{CT}$ (moist condition), although the responses of slopes to $RH_{CT}$ are not as significant as that with LTS. That is, under drier cloud top and stronger turbulence in clouds, positive correlations more likely occur, which are associated with stronger entrainment.

Figure 10 shows the distributions of sample number for LTS and $RH_{CT}$. It is clearly demonstrated that there exists a systemic difference in both LTS and $RH_{CT}$ between land and ocean; the frequencies of clouds under lower LTS (Fig. 10a) and $RH_{CT}$ (Fig. 10b) are higher over land than over ocean. The average LTS and $RH_{CT}$ over EC (10.3±2.3 and 67.8±14.8), EU (13.2±2.5 and 68.4±15.6), and WE (12.5±3.2 and 65.1±16.0) is generally lower than that over ECO (14.2±1.4 and 73.8±13.3), EUO (14.1±1.7 and 71.9±16.8), and WEO (17.9±3.1 and 75.8±23.5). Therefore, the opposite correlations over major industrial

regions and their adjacent oceans discussed in section 3.1 is probably associated with this systemic difference in the indicators of entrainment mixing, i.e., LTS and $RH_{CT}$. As shown in Fig. 9, positive correlations are more likely occur under the condition

of drier cloud top and stronger turbulence in clouds, which is more prevalent over land, while negative correlations normally occur under the condition of moister cloud top and weaker turbulence in clouds, which is common over ocean. The opposite CER-AOD/AI correlations over land and the negative one over ocean have also been generally found on a global scale (Grandey and Stier, 2010), but this is not always the case for some regions (Nakajima et al., 2001; Bréon et al., 2002), indicating that aerosol-cloud interactions remain challenging due to the interference of other influencing factors, such as different aerosol

microphysical properties and coincidentally changing meteorological conditions.

## 4 Summary

Using fourteen years of co-located aerosol and cloud observations from MODIS C6 L3 products, together with ERA-Interim reanalysis data, we systematically investigated correlations between cloud effective radius (CER) and aerosol optical depth (AOD) over three anthropogenic regions and correlations between CER and aerosol index (AI) over their adjacent oceans,

assessed the effects of satellite retrieval biases on the aerosol-cloud correlations, and then explored the underlying physical mechanisms. We also verified the reliability of the CER-AOD/AI correlation derived from satellite data by employing CERES Edition-4 L3 TOA albedo product.

Our analysis indicated that cloud effective radius is overall positively correlated with aerosol optical depth over land (positive slopes), but negatively correlated with aerosol index over oceans (negative slopes). Since CER is a function of both

AOD/AI and LWP, the assumption of constant LWP must be taken into consideration when estimating aerosol first indirect effect. The results shown that variations of LWP with AOD/AI in coarse LWP bins (40 g m$^{-2}$) can amplify the positive slope over land and the negative one over ocean. Therefore, we reanalyze the correlation between CER and AOD/AI for narrow intervals of LWP (5 g m$^{-2}$) in order to exclude the interference of the covariation of LWP and AOD/AI, and still get positive relationships over land and negative one over ocean.

The reliability of positive correlation between CER and AOD based on satellite remote sensing is still controversial (Zhao et al., 2018) due to the limitations of retrieval biases. Therefore, it is necessary to examine whether the positive correlation over land is real before exploring possible physical causes. We stratified data according to CF (with an interval of 0.04) and LWP (with an interval of 10 g m$^{-2}$) to isolate the Twomey effect. The results indicated that the changes in albedo at TOA corresponding to aerosol-induced changes in CER also lends credence to the reliability of positive correlation over land and

negative one over ocean.

Although the positive correlations between CER and AOD over land is believed to be a real relationship as discussed above, the magnitude of slope is still subject to artificial correlations due to the retrieval biases in both aerosol (cloud contamination and cloud adjacency effect) and cloud (partially cloudy and 3-D shaped clouds). To evaluate the contribution of retrieval biases on artificial correlations, we compared the correlations between CER and AOD/AI derived from PCL(partly cloudy) retrievals

and normal "overcast" retrievals, and took the impact of retrieval biases in aerosol into account by stratifying data according to CF. It is suggested that CER biases in PCL retrievals can induce an overestimation of the CER-AOD/AI slope, and the retrieval biases of aerosol can amplify the overestimation of slope caused by PCL retrievals. Additionally, our analysis shown that retrieval biases for 3-D shaped cloud also tend to result in a more positive slope of CER versus AOD/AI than its physically correct value. Therefore, the artificial positive correlation will be introduced if the following occurs: (a) PCL pixels are
erroneously identified as overcast pixels; (b) cloudy pixels are erroneously identified as clear; and (c) 3-D shaped clouds affect both cloud retrievals in their own cloud pixels and aerosol retrievals in surrounding cloud-free pixels.

We also explored potential physical mechanisms that can help to explain observed positive correlation between CER and AOD over land, including collision-coalescence and entrainment mixing. It is suggested that collision-coalescence seems not to be the dominant cause for positive slope of CER versus AOD over land, but there might exist a positive feedback that the
increased CER caused by increased aerosol might further increase CER by initializing collision-coalescence. Additionally, we stratified data according to the lower tropospheric stability (LTS) and relative humidity near cloud top ($RH_{CT}$) to differentiate environmental regimes, and found that the positive correlations more likely occur in case of drier cloud top and stronger turbulence in clouds, corresponding to stronger entrainment mixing. Furthermore, there exists systematic differences in LTS and $RH_{CT}$ between land dominated by the positive correlation and ocean dominated by the negative one, i.e., LTS and $RH_{CT}$
are lower over land than over ocean. Therefore, it is inferred that entrainment mixing might be a possible physical interpretation for such positive CER-AI slope.

It is acknowledged that although we have explored some uncertainties in clarifying the correlation between aerosols and clouds, the complexity of this issue and inherent limitations of polar-orbiting satellite measurements make it difficult to reach a definitive conclusion regarding causal relationships between CER and AOD/AI. Additional potential sources of the
uncertainties include:

- Satellite-derived cloud-top CER might deviate from cloud-base or whole cloud CER in case the clouds are non-adiabatic, and/or coagulation of cloud droplets occurs.

- As a vertical integral quantity, AOD/AI is not always a good proxy for cloud-base CCN concentrations in some cases (Stier, 2016).

- AOD/AI might be loosely related to cloud-base CCN if affected by deliquescence in high relative humidity environments, or if dust or other insoluble aerosol has a substantial share of the total aerosol.

- AOD/AI of clear-sky column retrievals might not be representative of the cloud-base CCN in the neighbouring cloudy pixels.

Unfortunately, it is quite difficult to evaluate these uncertainties in this study due to the limitations of the data. However,
our study still present sufficient evidence to warrant further investigations of the physical mechanisms using detailed in-situ field measurements of aerosol and cloud properties, especially vertical profiles.

*Data availability.* All data used in this study are publicly available. The MODIS/Aqua Level 3 Collection 6 datasets were acquired from the Level-1 and Atmosphere Archive & Distribution System (LAADS) Distributed Active Archive Center
(DAAC), located in the Goddard Space Flight Center in Greenbelt, Maryland (https://ladsweb.nascom.nasa.gov/). CERES SSF data were obtained from the NASA Langley Research Center Atmospheric Science Data Center (https://ceres.larc.nasa.gov/order_data.php). The ECMWF ERA-Interim data were collected from the ECMWF data server http://www.ecmwf.int/en/research/climate-reanalysis/.

*Author contributions.* HJ and XM designed the study and the statistical analysis. HJ processed the satellite data and drafted the manuscript. XM, JQ, YY, and TQ validated and debugged the results. All authors contributed to revising the manuscript.

*Competing interests.* The authors declare that they have no conflict of interest.

*Acknowledgements.* This study is supported by the National Natural Science Foundation of China grants (41475005 and 41675004) and the National Key R&D Program of China grants (2016YFA0600404). Yan Yin acknowledges funding support by the National Natural Science Foundation of China grants (41590873). We are grateful to the ease access to MODIS and CERES, provided by NASA. We also thank ECMWF for providing daily ERA-Interim reanalysis data in our work.

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

**Table 1. The differences between the slope of CER versus aerosol loading (AOD for land, AI for ocean) in PCL and in overcast retrievals in case of CF > 60% ($\Delta S_{|>60}$) and CF < 60% ($\Delta S_{|<60}$) under three LWP bins (g m$^{-2}$).**

| Region | 20 < LWP ≤ 60 | | 60 < LWP ≤ 100 | | 100 < LWP ≤ 140 | |
|---|---|---|---|---|---|---|
| | $\Delta S_{|<60}$ | $\Delta S_{|>60}$ | $\Delta S_{|<60}$ | $\Delta S_{|>60}$ | $\Delta S_{|<60}$ | $\Delta S_{|>60}$ |
| EC | 0.03 | 0.036 | 0.039 | 0.059 | 0.068 | 0.089 |
| EU | 0.05 | 0.069 | 0.023 | 0.046 | 0.014 | 0.047 |
| WE | 0.07 | 0.09 | 0.048 | 0.078 | 0.02 | 0.85 |
| ECO | 0.089 | 0.150 | 0.069 | 0.140 | 0.055 | 0.080 |
| EUO | 0.049 | 0.176 | 0.052 | 0.060 | 0.061 | 0.110 |
| WEO | 0.159 | 0.160 | 0.092 | 0.190 | 0.055 | 0.185 |

**Table 2. The range of LTS and RH$_{CT}$ for low, medium and high conditions, respectively.**

| Region | LTS (K) | | | RH$_{CT}$ (%) | | |
|---|---|---|---|---|---|---|
| | Low | Medium | High | Low | Medium | High |
| EC | 1.9 ~ 9.4 | 9.4 ~ 11.3 | 11.3 ~ 22.1 | 5.1 ~ 65.8 | 65.8 ~ 76.9 | 76.9 ~ 99.8 |
| EU | 5.8 ~ 12.1 | 12.1 ~ 14.1 | 14.1 ~ 26.3 | 4.7 ~ 64.4 | 64.4 ~ 76.4 | 76.4 ~ 99.8 |
| WE | 5.0 ~ 11.2 | 11.2 ~ 13.9 | 13.9 ~ 26.8 | 6.5 ~ 60.0 | 60.0 ~ 73.4 | 73.4 ~ 99.8 |
| ECO | 11.6 ~ 13.6 | 13.6 ~ 14.5 | 14.5 ~ 20.1 | 7.6 ~ 71.3 | 71.3 ~ 81.3 | 81.3 ~ 99.8 |
| EUO | 9.7 ~ 13.4 | 13.4 ~ 14.6 | 14.6 ~ 21.9 | 5.2 ~ 68.2 | 68.2 ~ 80.8 | 80.8 ~ 99.8 |
| WEO | 9.8 ~ 16.9 | 16.9 ~ 19.4 | 19.4 ~ 26.8 | 3.4 ~ 66.6 | 66.6 ~ 87.3 | 87.3 ~ 99.8 |

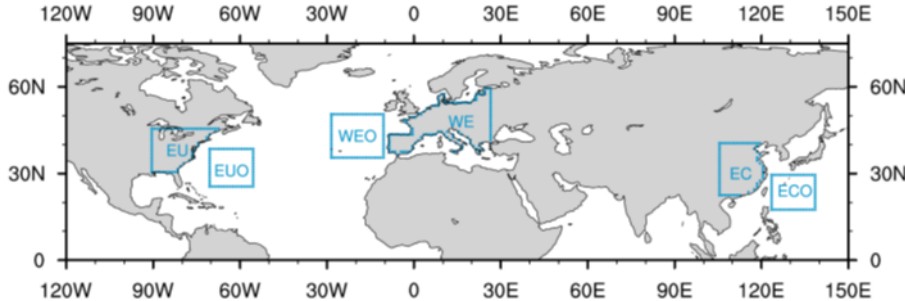

**Figure 1. Regions analyzed in this study: three anthropogenic regions (EC, EU, and WE) and their adjacent oceans (ECO, EUO, and WEO).**

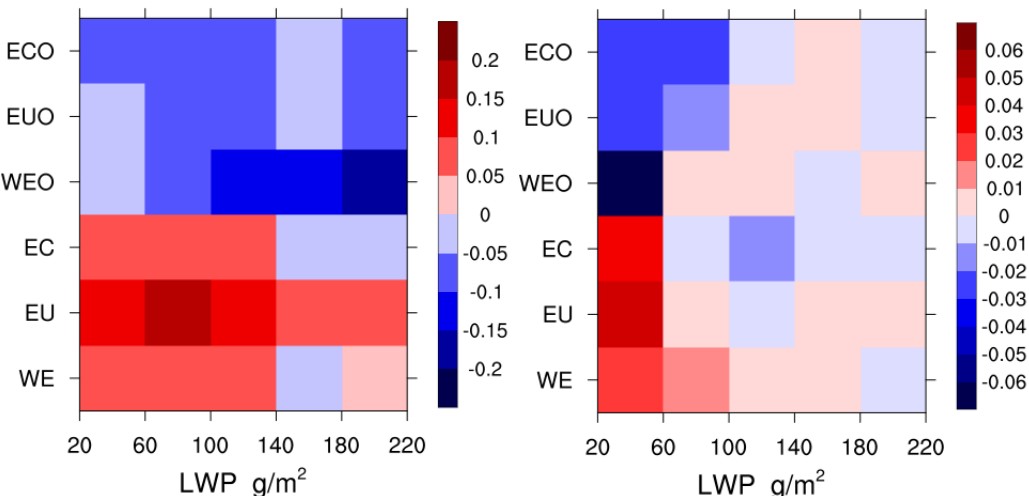

**Figure 2. The computed slopes of (a) CER versus aerosol loading (AOD for land, AI for ocean) (Ma et al., 2018) and (b) LWP versus aerosol loading on log-log scale over six regions, in which data are stratified according to LWP.**

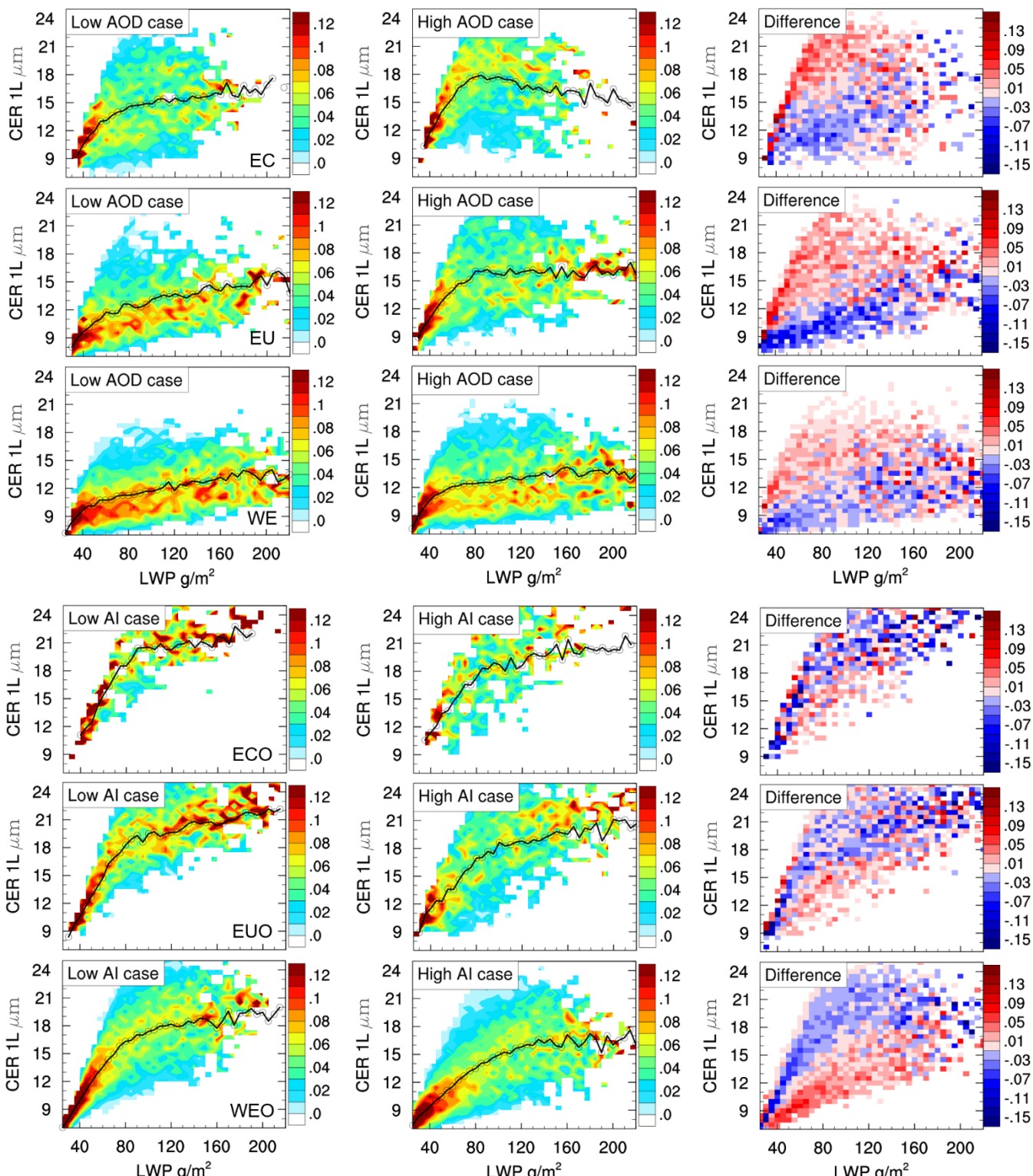

 **Figure 3. Joint histograms between LWP (x axis) and CER (y axis) for EC, EU, WE, ECO, EUO, and WEO. The first and second columns show the LWP-CER joint histograms for the low and high aerosol loading (AOD for land, AI for ocean) cases, respectively. The histograms are normalized so each column sums to 1, such that the histograms show the probability of observing a specific CER, given a certain LWP. The black line indicates the mean CER at each LWP. The third column shows the difference between the polluted and clean cases.**

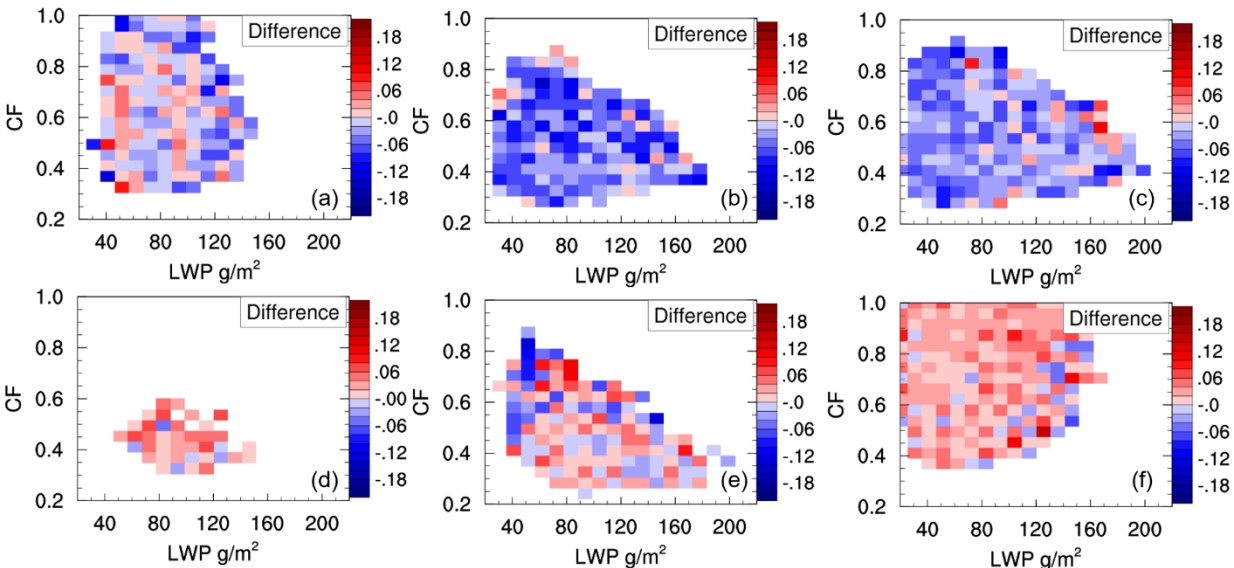

**Figure 4. The difference of TOA albedo for polluted (AOD/AI > 75th percentile) and clean case (AOD/AI < 25th percentile) (Δα) over (a) EC, (b) EU, (c) WE, (d) ECO, (e) EUO, and (f) WEO, in which data are stratified according to both LWP (x axis) and CF (y axis).**

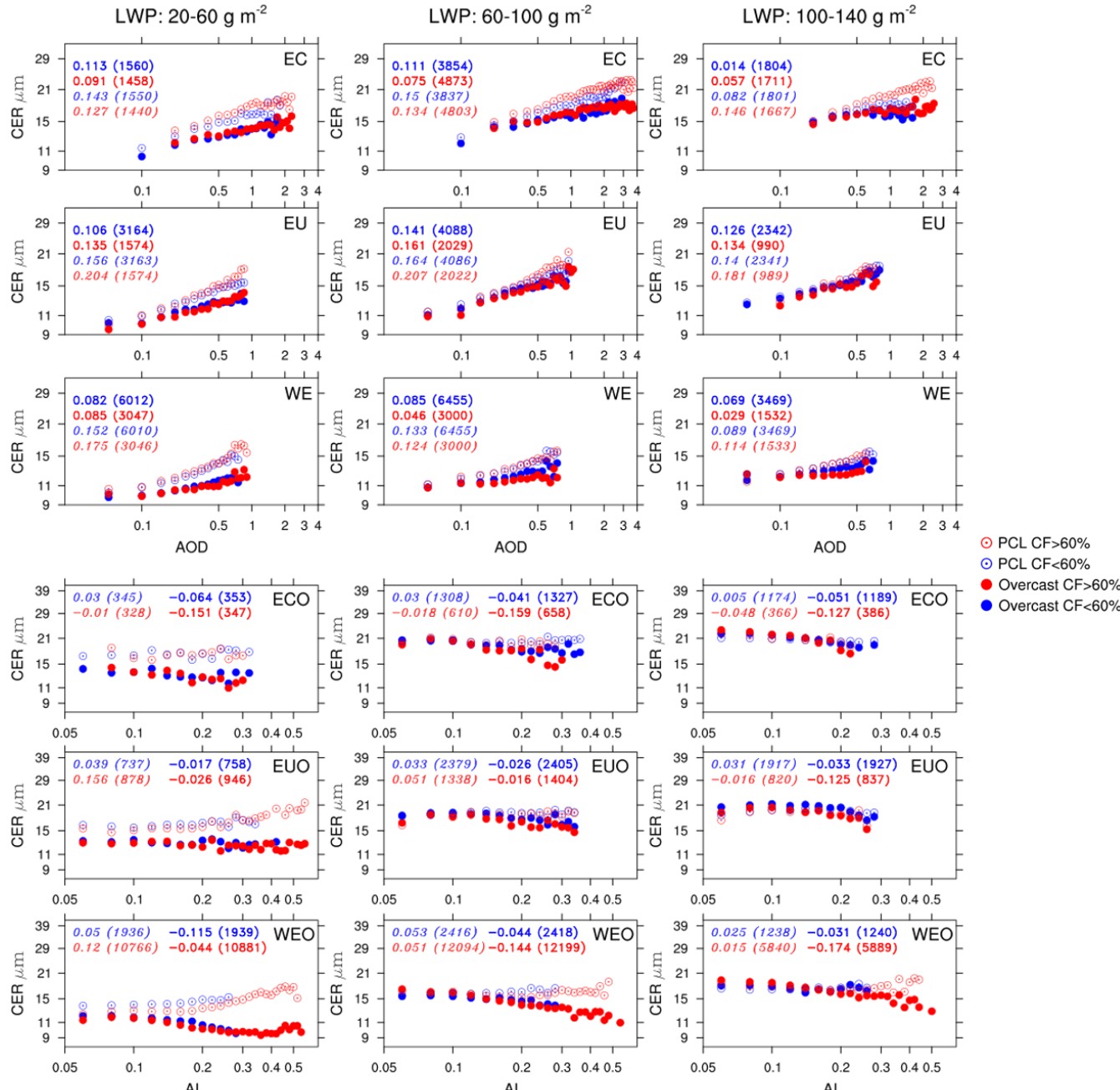

**Figure 5. Cloud effective radius (CER) varies with aerosol loading (AOD for land, AI for ocean) under three liquid water path (LWP) categories (20~60, 60~100, 100~140 g m⁻²). The figures from top to bottom show the CER-AOD/AI correlation over different regions (EC, EU, WE, ECO, EUO, WEO). The circles (dots) show the mean CER from PCL (overcast) retrieval at each AOD (AI) bin in case of CF > 60% (red) and CF < 60% (blue), respectively. The slopes on log-log scale along with the total number of samples**

**(bracketed) for these four categories are also provided in each panel, in which the numbers in italic (bold) represent PCL (overcast)**
740   **retrieval.**

745

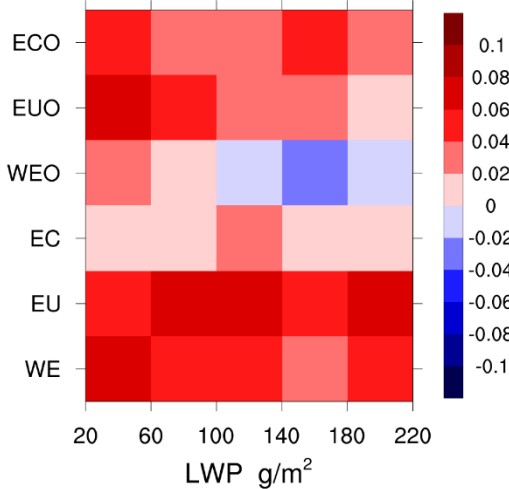

**Figure 6. The differences between CER$_{2.1\mu m}$-AOD (AI) slope and CER$_{3.7\mu m}$-AOD (AI) slope under all LWP bins over three anthropogenic regions (EC, EU, and WE) and their adjacent oceans (ECO, EUO, and WEO).**

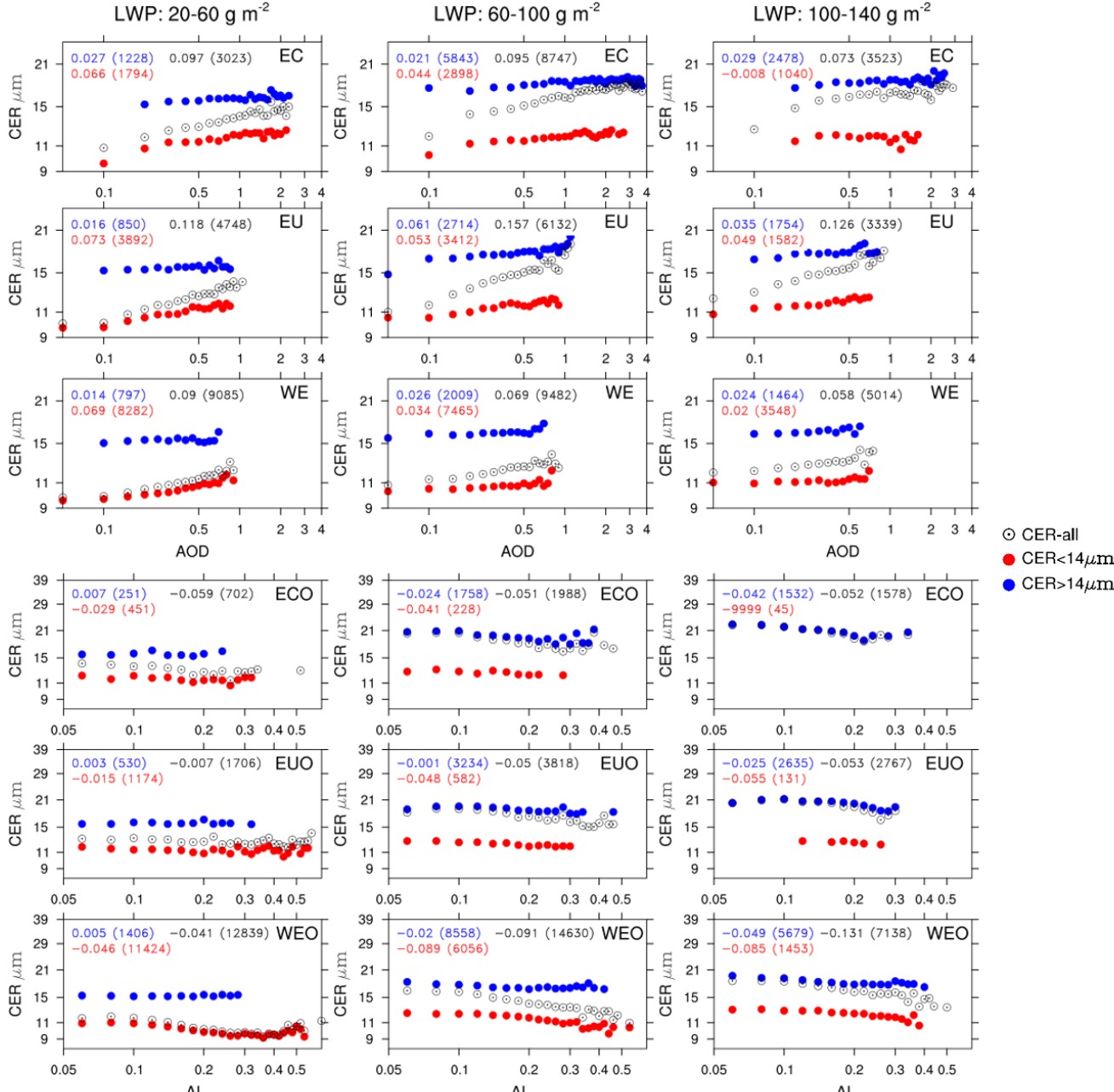

**Figure 7. Cloud effective radius (CER) varies with aerosol loading (AOD for land, AI for ocean) under three liquid water path (LWP) categories (20~60, 60~100, 100~140 g m$^{-2}$). The figures from top to bottom represent different regions (EC, EU, WE, ECO, EUO, and WEO). The blue (red) dots show the mean CER at each categories of AOD (AI) in case of CER > 14 µm (< 14 µm), and the black circles indicate the results that do not differentiate CER. The slopes on log-log scale along with the total number of samples (bracketed) for these three categories are also provided in each panel.**

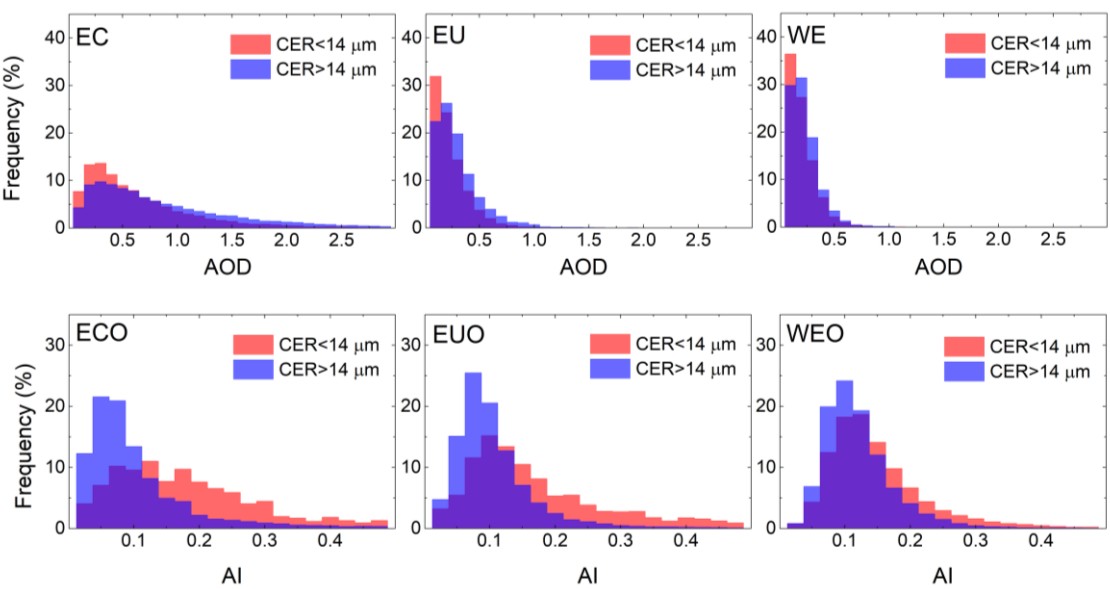

760   **Figure 8. Probability density functions of AOD (AI) for CER > 14 µm (blue) and CER < 14 µm (red) over three anthropogenic regions (EC, EU, and WE) and their adjacent oceans (ECO, EUO, and WEO).**

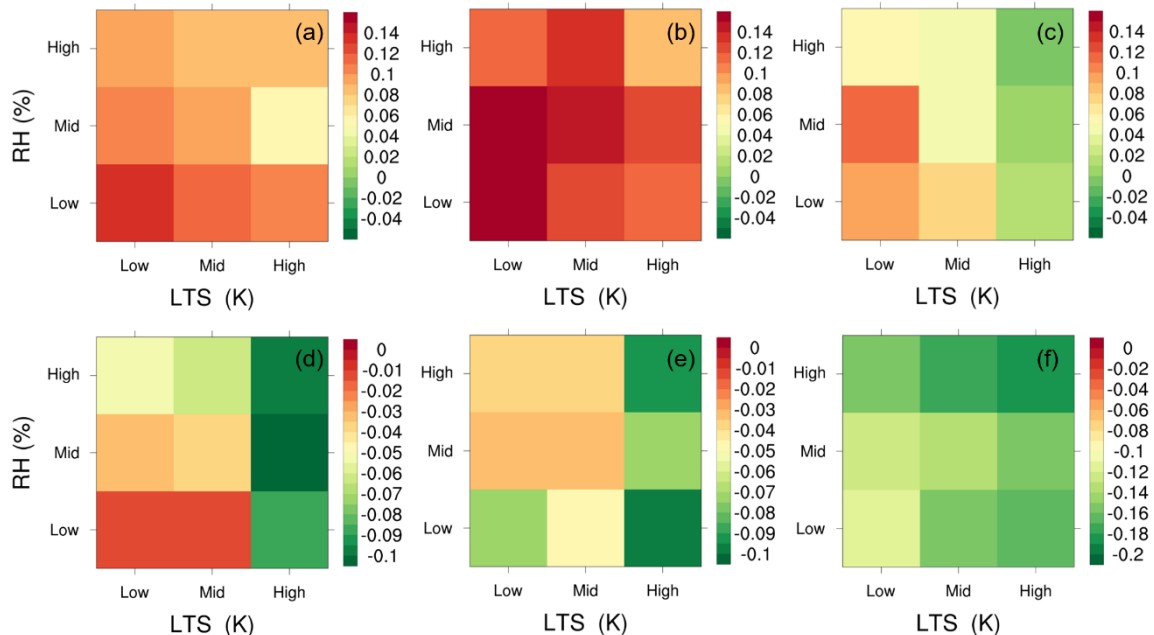

Figure 9. The slopes of CER versus AOD (AI) on log-log scale under low, medium, and high LTS and RH$_{CT}$ conditions, respectively, over (a) EC, (b) EU, (c) WE, (d) ECO, (e) EUO, and (f) WEO.

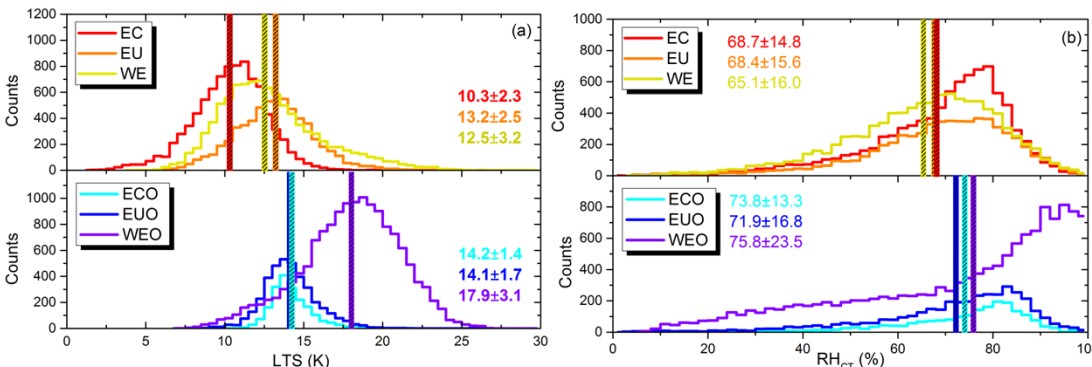

Figure 10. The distributions of sample number for (a) LTS and (b) RH$_{CT}$ over land (upper; EC, EU, and WE) and ocean (lower; ECO, EUO, and WEO). Vertical lines show the mean values. Texts with the corresponding color represent the mean and standard deviation for each region.