# Peer review of "Is Positive Correlation between Cloud Droplet Effective Radius and Aerosol Optical Depth over Land Due to Retrieval Artifacts or Real Physical Processes?"

_Atmospheric Chemistry and Physics, 2019_

## Referee Comment (RC1) · Anonymous Referee #1 · 8 Feb 2019

General comments:

The observed aerosol-cloud relationship from space remains controversial due to the wide range of influential factors, including the artifact retrieval of aerosol and cloud properties (large biases), and the confounding meteorological variables that simultaneously govern the aerosol and cloud systems. The manuscript is an useful attempt to address the controversial phenomenon frequently observed over high-polluted land—positive correlation between cloud droplet effective radius (CER) and aerosol loading. This study proposal new physical explanation (i.e., positive feedback caused by increased CER that in turn initialize the collision-coalescence processes) for this positive

correlation from satellite observations. This manuscript is logically organized, the analysis methods are technically sound but not novel, and the results are very interesting albeit some points not adequately illustrated. I have some comments on interpretation of the major results. Arguably, this topic is worth of further investigation. As such, I recommend its publication pending the following concerns satisfactorily addressed.

Major comments:

1. L29-38: The descriptions of aerosol climate effect (direct, indirect, ACI effects e.t.c. ) are duplicated in the 1st and 2nd paragraph. Thus, the authors can consider to combine them into one paragraph. 2. L50-69: these are about why you choose the proxy of aerosol index for CCN, and CER in the present study, which could be moved to section 2 and replaced with literature reviews of the role of vertical observations in the cloud-aerosol-precipitation interaction studies, which are omitted, including the measurements provided by CALIOP (Costantino and Bréon, ACP 2013; Zuidema, et al. BAMS 2016), Cloudsat (Christensen et al. JGR, 2016; Chen et al., JAS 2016; Peng J. et al., JAS), and TRMM (Wall et. al JAS 2014; Li et al., Rev Geophys, 2016; Guo et al., ACP 2018). Besides, the preference for use of aerosol index rather than AOD should be clarified in a more straightforward way, since there exist large uncertainties in the retrieval of Ångström exponent over land.

3. L75-81: The fourth factor can be added here impairing the quantification of aerosol-cloud interaction from observations: the vertical overlapping status of aerosol and cloud layers (e.g., Costantino and Bréon, ACP 2013, doi; 10.5194/acp-13-69-2013; Huang et al. JGR 2015, doi:10.1002/2014JD022898)

4. Section 2.2: The LTS is a proxy for the magnitude of the inversion strength in the lower troposphere. The readers are curious for the meaning with regard to the various LTS values. Large LTS means unstable conditions? Please clarify it. 5. L200-202: Any references to support the argument "Under cloudy sky, the response of . . . have larger retrieval biases"?? 6. In Fig. 4a, the samples of positive TOA albedo difference

is almost equal to the samples with negative albedo difference. Besides, Fig. 4b-c has a large fraction of positive albedo difference (more than 20%). How the authors claim that "implying that as AI increases, the reflected solar shortwave radiation at TOA will reduce over land while increase over ocean. " More importantly, necessary discussion is warranted for the difference of TOA albedo response to aerosol between over land and ocean. 7. Extensive previous studies have pointed to the saturation effect as the aerosol keeps rising. e.g, Breon et al. (Science 2002) argued that as the AI is greater than 0.15, the CER will keep constant. From Fig. 5 in this manuscript, most of the AI values in three regions over land are greater than 0.15. I wonder whether there exists such saturation effect when the samples are divided into two parts taking the threshold of AI=0.15? Or at the very least, the authors make sure to clarify the number of samples (the ratio) with the AI values less than 0.15 over each region of interest in this investigation. Also, the potential inference induced by saturation effect should be taken into account in the future submission of revision. 8. In section 3.3.2.2: Can "the degree of entrainment mixing" be represented by LTS? LTS is an indicator of temperature inversion proposed for stratocumulus over ocean. Further justification is needed. 9. Figure 9: why not show the The slopes of CER versus AI for the adjacent ocean areas? The readers are curious to know the difference of this slope between over land and ocean.

Minor comments: 1. L50: cloud droplet number -> cloud droplet number concentration, and make sure to correct for all instances in the manuscript. 2. L310-317: It is well known that much lower LTS (more unstable) and lower RH CT over land compared with over ocean, which is not probably concerned about any aerosol effect, but about the effect induced by the difference of underlying surface properties. Actually, even over ocean, positive correlation between CER and AI is observed over south Atlantic (e.g., in Fig. 2 of Nakajima et al. JGR 2001), and land (e.g., West US, West Africa in Fig. 2 of Breon et al. Science 2002), we can see negative slope between CER versus AI. It suggests that the relationship varies greatly by regions, and is still extremely challenging to be interpreted as a causal connection. I suggest the authors rephrase

the section more accurately. 3. What is the actual number of samples for each bin in Figures 4-7? Clarification in the figure caption will help the readers to better follow. 4. What does it mean for the labelled number in different color in each panel in Fig. 5? The slope? It is better to clarify in figure caption.
* * *

---

## Referee Comment (RC2) · Anonymous Referee #2 · 11 Apr 2019

Before getting to the review of the manuscript, I want to say. . . why do people accept papers to review and then not review them (and don't let the editor know this quickly)? It seems like this happens way too often, as it did here.

Ok, now to the manuscript...

This manuscript explores the differences between land and ocean regions in the correlation of near-cloud aerosol index (AI) to cloud effective radius (CER) and liquid water path (LWP). The paper looks various potential reasons (real and artificial) for the positive correlations between AI and CER are over land, as opposed to the more-expected negative correlation over oceans. While no precise reasons are found, several hypothe-

ses were able to be eliminated. I feel that the paper may be useful, though I have one major concern and several minor comments that I feel need to be addressed before publication.

Major comment:

The Angstrom exponent, used for deriving the aerosol index, is notoriously bad over land for MODIS Dark Target land retrieval (not sure about Deep Blue retrievals, and I'm unsure about which retrieval products were used here, though I believe that the 3 land regions used in this paper will mostly be Dark Target rather than Deep Blue). In the Levy et al. 2013 paper describing collection 6: "On a global basis, we and others have found little quantitative skill in MODIS-retrieved aerosol size parameters over land (e.g., Levy et al., 2010; Mielonen et al., 2011). We have decided to discontinue further attempts at validating Ångström Exponent (AE) and fine-AOD. A user can still choose to derive AE (from spectral AOD) or fine-AOD (from product of $\tau$ $\eta$) and evaluate the results themselves." In Levy et al. 2010, they show that the R^2 of the Angstrom exponent over land is only 0.3 with AERONET.

Levy et al. 2013: https://www.atmos-meas-tech.net/6/2989/2013/amt-6-2989-2013.pdf
Levy et al. 2010: https://www.atmos-chem-phys.net/10/10399/2010/acp-10-10399-2010.pdf

Related, for research on estimating surface-level PM2.5 from AOD, it would be useful to use the Angstrom exponent or fine-mode AOD products to help filter coarse PM from the PM2.5 estimate. However, due to the poor ability for MODIS to retrieve these products over land, the AOD-PM2.5 community uses just AOD as using the Angstrom exponent or fine-mode products adds noise to evaluations. It is unclear why the aerosol-cloud interactions community should have more confidence using the MODIS Angstrom exponent over land.

It seems that the aerosol index over land used here may be very poor (due to the Angstrom exponent uncertainty_, and this could be contributing to counterintuitive results over land. There is no mention of this in the paper. I feel that the authors should evaluate their aerosol index data in their 3 regions against AERONET to show if their aerosol index product has any skill in their land regions, and discussion needs to be added.

Minor comments

L29: "Earth" should be "Earth's"

L31-33: It is more precise to say that ACIs are the largest forcing uncertainty to present climate change. I'm nearly certain that the uncertainty in what humans will do (e.g. in terms of how much CO2 we will emit and other changes) is the biggest uncertainty in the anthropogenic contribution to *future* climate change.

L50-51: Kohler theory only describes the relationship between a particle and its critical supersaturation for activation. Cloud droplet number at the cloud base depends on the number distribution of critical supersaturations and the updraft velocity. The second part of the sentence discusses these effects, but I feel that this sentence can be rewritten to be more precise.

L64: "Loosely related". Is it possible to be more quantitative here (e.g. using numbers from the Stier reference)?

L68: Same, can you be quantitative?

L125: Which MODIS AOD retrieval algorithms are you using (dark target, deep blue, MAIAC)?

L268: "Become*s*"

L269: Either "*the* collision-coalescence process" or "collision-coalescence process*es*".

Figure 7 and associated discussion: It is unsurprising that the slope of CER-AI is stronger when including all data vs. when binning to >14um and <14um. If the slope

is positive, more observations will move from the <14um to >14um with increasing AI. With increasing AI, a CER of 13.9 um (which would be at the large end of the <14 um bin) would shift to 14.1 um (which would be on the low end of the >14 um bin), so increasing AI removes some of the higher <14um cases and creates new lower >14 um cases... the all prevents the CER for the two size bins from change much with AI. Hence, the slopes for the binned sizes are buffered from changes with AI. A similar phenomena will occur of a negative slope. Thus, the discussion in lines 274-278 seems unnecessary.

---

## Author Comment (AC1) · 8 May 2019

Comments are in black and responses are in blue.

**General comments:**

The observed aerosol-cloud relationship from space remains controversial due to the wide range of influential factors, including the artifact retrieval of aerosol and cloud properties (large biases), and the confounding meteorological variables that simultaneously govern the aerosol and cloud systems. The manuscript is an useful attempt to address the controversial phenomenon frequently observed over high-polluted land— ˇpositive correlation between cloud droplet effective radius (CER) and aerosol loading. This study proposal new physical explanation (i.e., positive feedback caused by increased CER that in turn initialize the collision-coalescence processes) for this positive correlation from satellite observations. This manuscript is logically organized, the analysis methods are technically sound but not novel, and the results are very interesting albeit some points not adequately illustrated. I have some comments on interpretation of the major results. Arguably, this topic is worth of further investigation. As such, I recommend its publication pending the following concerns satisfactorily addressed.

We thank the reviewer for taking the time to assess the manuscript and for providing helpful comments and suggestions to improve the manuscript. We have revised the manuscript carefully according to the reviewer's comments. Please see the following detailed point-by-point responses.

**Major comments:**

1. L29-38: The descriptions of aerosol climate effect (direct, indirect, ACI effects e.t.c.) are duplicated in the 1st and 2nd paragraph. Thus, the authors can consider to combine them into one paragraph.

   Thanks for the suggestion. We have combined them into one paragraph in the revised manuscript as suggested.

2. L50-69: these are about why you choose the proxy of aerosol index for CCN, and CER in the present study, which could be moved to section 2 and replaced with literature reviews of the role of vertical observations in the cloud-aerosol-precipitation interaction studies, which are omitted, including the measurements provided by CALIOP (Costantino and Bréon, ACP 2013; Zuidema, et al. BAMS 2016), Cloudsat (Christensen et al. JGR, 2016; Chen et al., JAS 2016; Peng J. et al., JAS), and TRMM (Wall et. al JAS 2014; Li et al., Rev Geophys, 2016; Guo et al., ACP 2018).

   We are grateful for the important references provided by the reviewer. As suggested, L50-69 have been moved to section 2, and the literature reviews of the vertical observations of aerosols, clouds, and precipitation have been included in the revised manuscript.

   Besides, the preference for use of aerosol index rather than AOD should be clarified in a more straightforward way, since there exist large uncertainties in the retrieval of Ångström exponent over land.

   Thanks for the comment. We acknowledge that the MODIS retrieval of Ångström exponent over land may be problematic. The discussions related to the uncertainties in the retrieval of AE have

been added to section 2.1. To eliminate the possible interference of AE retrieval accuracy on the estimation of aerosol-cloud correlations, we conduct a similar statistical analysis by using AOD, rather than AI, as an aerosol proxy, and found that the results only change slightly, but not influence our conclusions. In the revised manuscript, the results of using AI as the proxy over land have been replaced by that of using AOD, and the text has been revised accordingly.

3. L75-81: The fourth factor can be added here impairing the quantification of aerosol cloud interaction from observations: the vertical overlapping status of aerosol and cloud layers (e.g., Costantino and Bréon, ACP 2013, doi; 10.5194/acp-13-69-2013; Huang et al. JGR 2015, doi:10.1002/2014JD022898)

Thanks for the suggestion. The vertical overlapping status of aerosol and cloud layers has been added as fourth factor.

4. Section 2.2: The LTS is a proxy for the magnitude of the inversion strength in the lower troposphere. The readers are curious for the meaning with regard to the various LTS values. Large LTS means unstable conditions? Please clarify it.

Thanks for the reminder. Large LTS is associated with stable conditions in which vertical mixing is suppressed. We have clarified the meaning of the LTS in section 2.2.

5. L200-202: Any references to support the argument "Under cloudy sky, the response of … have larger retrieval biases"??

Thanks for the reminder. Relevant references have been cited here.

6. In Fig. 4a, the samples of positive TOA albedo difference is almost equal to the samples with negative albedo difference. Besides, Fig. 4b-c has a large fraction of positive albedo difference (more than 20%). How the authors claim that "implying that as AI increases, the reflected solar shortwave radiation at TOA will reduce over land while increase over ocean." More importantly, necessary discussion is warranted for the difference of TOA albedo response to aerosol between over land and ocean.

We calculated the ratio of samples with positive values to total samples over the six regions in Fig. 4, and found that the ratio over EC, EU, and WE is 35 %, 16 %, and 19 %, respectively, which is evidently less than that over ECO (85 %), EUO (61 %), and WEO (84 %). Overall, the negative TOA albedo difference dominates over land whereas the positive one dominates over ocean, although a relatively large ratio over land (35 % over EC) and a relatively small one over ocean (61 % over EUO) are found. The sentence "*implying that as AI increases….over land while increase over ocean*" has been rephrased as, "*This result implies that as AOD/AI increases, the reflected solar shortwave radiation at TOA generally tends to decrease over land but increase over ocean*". The discussion associated with the difference of TOA albedo response to aerosol between over land and ocean, has been included in the revised manuscript (line 276-279), as suggested by the reviewer.

7. Extensive previous studies have pointed to the saturation effect as the aerosol keeps rising. e.g, Breon et al. (Science 2002) argued that as the AI is greater than 0.15, the CER will keep constant. From Fig. 5 in this manuscript, most of the AI values in three regions over land are greater than 0.15. I wonder whether there exists such saturation effect when the samples are divided into two

parts taking the threshold of AI=0.15? Or at the very least, the authors make sure to clarify the number of samples (the ratio) with the AI values less than 0.15 over each region of interest in this investigation. Also, the potential inference induced by saturation effect should be taken into account in the future submission of revision.

Thanks for these suggestions. The number of samples (the ratio) with AI values less than 0.15 over six regions is now shown as suggested by the reviewer (Figure R1). For regions over land, the number of samples with AI less than 0.15 is only a small part of the total number of samples. However, for regions over ocean, the ratio can be as high as 60% to 70%.

To further investigate the saturation effect, we averaged CER in a smaller AOD/AI interval of 0.02 (Fig. S1). The results show that the saturation effect was found in three ocean regions (ECO, EUO, and WEO) adjacent to the anthropogenic regions, but with a higher threshold (AI = 0.3) than that derived from the analysis on global oceans (Breon et al., 2002). However, it is noted that no saturation effect was found in the regions with strong anthropogenic emissions over land (EC, EU, and WE), i.e., CER shows a continuous increase as AOD increases (Fig. S1), suggesting that the saturation effects does not appear to interfere with the observed positive slopes over land. According to the reviewer's suggestions, the results and discussions related to saturation effects have been inserted in in section 3.1 (line 237-246 in the revised manuscript).

[Figure]

**Figure R1. Probability density functions of AI over (a) EC, (b) EU, (c) WE, (d) ECO, (e) EUO, and (f) WEO. Each panel provides the number of samples with AI less than 0.15 (num) and the ratio of num to total number of samples (ratio).**

[Figure]

**Figure S1.** Cloud effective radius (CER) as a function of aerosol optical depth (AOD) over (a) EC, and (b) EU, (c) WE, and aerosol index (AI) over (d) ECO, (e) EUO, and (f) WEO. The dots show the mean CER at each AOD/AI bin of 0.02. The slopes on log-log scale and the least-square fits for AI < 0.3 (blue) and AI > 0.3 (red), respectively, are provided in panel (d), (e), and (f).

8. In section 3.3.2.2: Can "the degree of entrainment mixing" be represented by LTS? LTS is an indicator of temperature inversion proposed for stratocumulus over ocean. Further justification is needed.

   The entrainment rate in stratocumulus clouds was found to be larger for the smaller LTS (Dal Gesso et al., 2014). Although LTS was initially proposed as relevant for marine stratocumulus clouds (Klein and Hartmann, 1993), it has also been widely used to characterize atmospheric stability over both land and ocean, and thereby indicating the vertical mixing (Chen et al., 2014; Chen et al., 2016; Liu et al., 2017; Ma et al., 2018). As stated by De Rooy et al. (2013), the degree of entrainment mixing is controlled by both the intensity of turbulence in clouds and the relative humidity of air outside clouds. Here, LTS is thus employed as an indicator of the intensity of turbulence in liquid clouds, which can be represented by the stability of atmosphere where the clouds occurs, i.e., lower troposphere. The associated discussions have been included in section 3.3.2.2 (line 354-358 in the revised manuscript).

9. Figure 9: why not show the The slopes of CER versus AI for the adjacent ocean areas? The readers are curious to know the difference of this slope between over land and ocean

   Thanks for the reminder. The slopes of CER versus AI under low, medium, and high LTS and $RH_{CT}$ conditions over ocean have been included in the revised manuscript.

**Minor comments:**

1. L50: cloud droplet number -> cloud droplet number concentration, and make sure to correct for all instances in the manuscript.

   Corrected.

2. L310-317: It is well known that much lower LTS (more unstable) and lower RH CT over land compared with over ocean, which is not probably concerned about any aerosol effect, but about

the effect induced by the difference of underlying surface properties. Actually, even over ocean, positive correlation between CER and AI is observed over south Atlantic (e.g., in Fig. 2 of Nakajima et al. JGR 2001), and land (e.g., West US, West Africa in Fig. 2 of Breon et al. Science 2002), we can see negative slope between CER versus AI. It suggests that the relationship varies greatly by regions, and is still extremely challenging to be interpreted as a causal connection. I suggest the authors rase the section more accurately.

We agree with the reviewer that the systemic difference of LTS and $RH_{CT}$ between land and ocean is induced by the different underlying surface properties rather than aerosol effects. The analysis presented here is to illustrate that the difference in LTS and $RH_{CT}$ is a possible cause for the opposite CER-AOD/AI correlations over land and ocean shown in section 3.1. The positive correlations are more likely to occur under the condition of drier cloud top and stronger turbulence in clouds, which is more prevalent over land, while the negative one normally occur under the condition of moister cloud top and weaker turbulence in clouds, which is common over ocean.

We agree that this relationship varies greatly by regions as stated by the reviewer, though the positive CER-AOD/AI correlations over land and the negative one over ocean have been generally found on a global scale (Grandey and Stier, 2010). This inconsistency indicates that aerosol-cloud interactions remain challenging due to the interference of other influencing factors, such as different aerosol microphysical properties and coincidentally changing meteorological conditions.

According the reviewer's suggestion, we have rephrased this part in the revised manuscript.

3. What is the actual number of samples for each bin in Figures 4-7? Clarification in the figure caption will help the readers to better follow.

Thanks for these suggestions. For Figures 5 and 7, the number of samples for each category has been added to the figure. The number of samples for each bin in Figures 4 and 6 has also been shown in Fig. S2 and Fig. S3.

4. What does it mean for the labelled number in different color in each panel in Fig. 5? The slope? It is better to clarify in figure caption.

Thanks for the reminder. The labelled number in different color represents the slope on log-log scale. We have clarified in figure caption as suggested.

[revised manuscript text omitted]

**Figure List**

**Figure S1.** Cloud effective radius (CER) as a function of aerosol optical depth (AOD) over (a) EC, and (b) EU, (c) WE, and aerosol index (AI) over (d) ECO, (e) EUO, and (f) WEO. The dots show the mean CER at each AOD/AI bin of 0.02. The slopes on log-log scale and the least-square fits for AI < 0.3 (blue) and AI > 0.3 (red), respectively, are provided in panel (d), (e), and (f).

**Figure S2.** The number of samples for each LWP (x axis) and CF (y axis) bin over (a) EC, (b) EU, (c) WE, (d) ECO, (e) EUO, and (f) WEO.

**Figure S3.** The number of samples for each LWP bin over (a) EC, (b) EU, (c) WE, (d) ECO, (e) EUO, and (f) WEO.

**Figure S1**

[Figure]

**Figure S2**

[Figure]

**Figure S3**

[Figure]

---

## Author Comment (AC2) · 8 May 2019

Comments are in black and responses are in blue.

This manuscript explores the differences between land and ocean regions in the correlation of near-cloud aerosol index (AI) to cloud effective radius (CER) and liquid water path (LWP). The paper looks various potential reasons (real and artificial) for the positive correlations between AI and CER are over land, as opposed to the more-expected negative correlation over oceans. While no precise reasons are found, several hypotheses were able to be eliminated. I feel that the paper may be useful, though I have one major concern and several minor comments that I feel need to be addressed before publication.

Thanks to the reviewer for very helpful comments and suggestions, which have allowed us to clarify and improve the manuscript. We have revised the manuscript carefully according to the reviewer's comments. At the same time, we are grateful for the important references provided by the reviewer. Please see the following detailed point-by-point responses.

**Major comment:**

1. The Angstrom exponent, used for deriving the aerosol index, is notoriously bad over land for MODIS Dark Target land retrieval (not sure about Deep Blue retrievals, and I'm unsure about which retrieval products were used here, though I believe that the 3 land regions used in this paper will mostly be Dark Target rather than Deep Blue). In the Levy et al. 2013 paper describing collection 6: "On a global basis, we and others have found little quantitative skill in MODIS-retrieved aerosol size parameters over land (e.g., Levy et al., 2010; Mielonen et al., 2011). We have decided to discontinue further attempts at validating Ångström Exponent (AE) and fine-AOD. A user can still choose to derive AE (from spectral AOD) or fine-AOD (from product of $\tau$ $\eta$) and evaluate the results themselves." In Levy et al. 2010, they show that the R^2 of the Angstrom exponent over land is only 0.3 with AERONET.

   Levy et al. 2013: https://www.atmos-meas-tech.net/6/2989/2013/amt-6-2989-2013.pdf

   Levy et al. 2010: https://www.atmos-chem-phys.net/10/10399/2010/acp-10-10399-2010.pdf

   Related, for research on estimating surface-level PM2.5 from AOD, it would be useful to use the Angstrom exponent or fine-mode AOD products to help filter coarse PM from the PM2.5 estimate. However, due to the poor ability for MODIS to retrieve these products over land, the AOD-PM2.5 community uses just AOD as using the Angstrom exponent or fine-mode products adds noise to evaluations. It is unclear why the aerosol-cloud interactions community should have more confidence using the MODIS Angstrom exponent over land.

   It seems that the aerosol index over land used here may be very poor (due to the Angstrom exponent uncertainty_, and this could be contributing to counterintuitive results over land. There is no mention of this in the paper. I feel that the authors should evaluate their aerosol index data in their 3 regions against AERONET to show if their aerosol index product has any skill in their land regions, and discussion needs to be added.

   Thanks very much for valuable suggestions. In this study, the AOD products retrieved by the Dark

Target (DT) algorithm are used, which has been clarified in the revised manuscript. We agree with the reviewer that there exist large uncertainties in the MODIS retrieval of Ångström exponent over land. The associated discussions have been added to section 2.1 as suggested by the reviewer. To eliminate the possible interference of AE retrieval accuracy on the estimation of aerosol-cloud correlations, we conduct a similar statistical analysis by using AOD, rather than AI, as an aerosol proxy over land, which has been validated extensively (Remer et al., 2005; Tripathi et al., 2005). It is found that the results only change slightly, but not influence our conclusions, which indicates that the uncertainties in AI does not contribute to the positive correlation between CER and AI over land. In revised manuscript, the results of using AI as the proxy over land have been replaced by that of using AOD, and the text has been revised accordingly.

**Minor comments**

1. L29: "Earth" should be "Earth's"

   Corrected.

2. L31-33: It is more precise to say that ACIs are the largest forcing uncertainty to present climate change. I'm nearly certain that the uncertainty in what humans will do (e.g. in terms of how much CO2 we will emit and other changes) is the biggest uncertainty in the anthropogenic contribution to *future* climate change.

   Thanks for the reminder. We have revised "future" to "present" in the revised manuscript.

3. L50-51: Kohler theory only describes the relationship between a particle and its critical supersaturation for activation. Cloud droplet number at the cloud base depends on the number distribution of critical supersaturations and the updraft velocity. The second part of the sentence discusses these effects, but I feel that this sentence can be rewritten to be more precise.

   Thanks for the suggestion. We have rephrased the sentence in the revised manuscript as follows:

   *"The relationship between aerosol and cloud droplet number, $N_d$, at cloud base fundamentally is controlled by both the critical supersaturation of aerosol particles, which described by Köhler theory, and the maximum supersaturation at cloud base, which determined by the updraft speed and the competition of the existing CCN for the available humidity."*

4. L64: "Loosely related". Is it possible to be more quantitative here (e.g. using numbers from the Stier reference)?

   Thanks for the suggestion. We have include the below quantitative description here:

   *"Stier (2016) stated that 52 % of the area of the globe shows correlation coefficients between $CCN_{0.2\%}$ at cloud base and AI below 0.5."*

5. L68: Same, can you be quantitative?

   Unfortunately, it is difficult at the moment to give a quantitative correlation between cloud base CCN and AI associated with each individual cause mentioned here. Only the overall quantitative description is able to be given (please see last response).

6. L125: Which MODIS AOD retrieval algorithms are you using (dark target, deep blue, MAIAC)?

We use the Dark Target AOD in this study, which has been clarified in the revised manuscript.

7. L268: "Become*s*"

Corrected.

8. L269: Either "*the* collision-coalescence process" or "collision-coalescence process*es*".

Corrected.

9. Figure 7 and associated discussion: It is unsurprising that the slope of CER-AI is stronger when including all data vs. when binning to >14um and <14um. If the slope is positive, more observations will move from the <14um to >14um with increasing AI. With increasing AI, a CER of 13.9 um (which would be at the large end of the <14 um bin) would shift to 14.1 um (which would be on the low end of the >14 um bin), so increasing AI removes some of the higher <14um cases and creates new lower >14 um cases… the all prevents the CER for the two size bins from change much with AI. Hence, the slopes for the binned sizes are buffered from changes with AI. A similar phenomena will occur of a negative slope. Thus, the discussion in lines 274-278 seems unnecessary.

Thanks for this hint. We removed this statement in the revised manuscript which might be too hasty.

**References**

[revised manuscript text omitted]

**Figure List**

**Figure S1.** Cloud effective radius (CER) as a function of aerosol optical depth (AOD) over (a) EC, and (b) EU, (c) WE, and aerosol index (AI) over (d) ECO, (e) EUO, and (f) WEO. The dots show the mean CER at each AOD/AI bin of 0.02. The slopes on log-log scale and the least-square fits for AI < 0.3 (blue) and AI > 0.3 (red), respectively, are provided in panel (d), (e), and (f).

**Figure S2.** The number of samples for each LWP (x axis) and CF (y axis) bin over (a) EC, (b) EU, (c) WE, (d) ECO, (e) EUO, and (f) WEO.

**Figure S3.** The number of samples for each LWP bin over (a) EC, (b) EU, (c) WE, (d) ECO, (e) EUO, and (f) WEO.

**Figure S1**

[Figure]

**Figure S2**

[Figure]

**Figure S3**

[Figure]

---

## Editor Decision (ED1)

General comments:

The observed aerosol-cloud relationship from space remains controversial due to the wide range of influential factors, including the artifact retrieval of aerosol and cloud properties (large biases), and the confounding meteorological variables that simultaneously govern the aerosol and cloud systems. The manuscript is an useful attempt to address the controversial phenomenon frequently observed over high-polluted land—positive correlation between cloud droplet effective radius (CER) and aerosol loading. This study proposal new physical explanation (i.e., positive feedback caused by increased CER that in turn initialize the collision-coalescence processes) for this positive

correlation from satellite observations. This manuscript is logically organized, the analysis methods are technically sound but not novel, and the results are very interesting albeit some points not adequately illustrated. I have some comments on interpretation of the major results. Arguably, this topic is worth of further investigation. As such, I recommend its publication pending the following concerns satisfactorily addressed.

Major comments:

1. L29-38: The descriptions of aerosol climate effect (direct, indirect, ACI effects e.t.c. ) are duplicated in the 1st and 2nd paragraph. Thus, the authors can consider to combine them into one paragraph. 2. L50-69: these are about why you choose the proxy of aerosol index for CCN, and CER in the present study, which could be moved to section 2 and replaced with literature reviews of the role of vertical observations in the cloud-aerosol-precipitation interaction studies, which are omitted, including the measurements provided by CALIOP (Costantino and Bréon, ACP 2013; Zuidema, et al. BAMS 2016), Cloudsat (Christensen et al. JGR, 2016; Chen et al., JAS 2016; Peng J. et al., JAS), and TRMM (Wall et. al JAS 2014; Li et al., Rev Geophys, 2016; Guo et al., ACP 2018). Besides, the preference for use of aerosol index rather than AOD should be clarified in a more straightforward way, since there exist large uncertainties in the retrieval of Ångström exponent over land.

3. L75-81: The fourth factor can be added here impairing the quantification of aerosol-cloud interaction from observations: the vertical overlapping status of aerosol and cloud layers (e.g., Costantino and Bréon, ACP 2013, doi; 10.5194/acp-13-69-2013; Huang et al. JGR 2015, doi:10.1002/2014JD022898)

4. Section 2.2: The LTS is a proxy for the magnitude of the inversion strength in the lower troposphere. The readers are curious for the meaning with regard to the various LTS values. Large LTS means unstable conditions? Please clarify it. 5. L200-202: Any references to support the argument "Under cloudy sky, the response of . . . have larger retrieval biases"?? 6. In Fig. 4a, the samples of positive TOA albedo difference

is almost equal to the samples with negative albedo difference. Besides, Fig. 4b-c has a large fraction of positive albedo difference (more than 20%). How the authors claim that "implying that as AI increases, the reflected solar shortwave radiation at TOA will reduce over land while increase over ocean. " More importantly, necessary discussion is warranted for the difference of TOA albedo response to aerosol between over land and ocean. 7. Extensive previous studies have pointed to the saturation effect as the aerosol keeps rising. e.g, Breon et al. (Science 2002) argued that as the AI is greater than 0.15, the CER will keep constant. From Fig. 5 in this manuscript, most of the AI values in three regions over land are greater than 0.15. I wonder whether there exists such saturation effect when the samples are divided into two parts taking the threshold of AI=0.15? Or at the very least, the authors make sure to clarify the number of samples (the ratio) with the AI values less than 0.15 over each region of interest in this investigation. Also, the potential inference induced by saturation effect should be taken into account in the future submission of revision. 8. In section 3.3.2.2: Can "the degree of entrainment mixing" be represented by LTS? LTS is an indicator of temperature inversion proposed for stratocumulus over ocean. Further justification is needed. 9. Figure 9: why not show the The slopes of CER versus AI for the adjacent ocean areas? The readers are curious to know the difference of this slope between over land and ocean.

Minor comments: 1. L50: cloud droplet number -> cloud droplet number concentration, and make sure to correct for all instances in the manuscript. 2. L310-317: It is well known that much lower LTS (more unstable) and lower RH CT over land compared with over ocean, which is not probably concerned about any aerosol effect, but about the effect induced by the difference of underlying surface properties. Actually, even over ocean, positive correlation between CER and AI is observed over south Atlantic (e.g., in Fig. 2 of Nakajima et al. JGR 2001), and land (e.g., West US, West Africa in Fig. 2 of Breon et al. Science 2002), we can see negative slope between CER versus AI. It suggests that the relationship varies greatly by regions, and is still extremely challenging to be interpreted as a causal connection. I suggest the authors rephrase

the section more accurately. 3. What is the actual number of samples for each bin in Figures 4-7? Clarification in the figure caption will help the readers to better follow. 4. What does it mean for the labelled number in different color in each panel in Fig. 5? The slope? It is better to clarify in figure caption.
* * *
[Figure]

[Figure]

Before getting to the review of the manuscript, I want to say... why do people accept papers to review and then not review them (and don't let the editor know this quickly)? It seems like this happens way too often, as it did here.

Ok, now to the manuscript...

This manuscript explores the differences between land and ocean regions in the correlation of near-cloud aerosol index (AI) to cloud effective radius (CER) and liquid water path (LWP). The paper looks various potential reasons (real and artificial) for the positive correlations between AI and CER are over land, as opposed to the more-expected negative correlation over oceans. While no precise reasons are found, several hypothe-
ses were able to be eliminated. I feel that the paper may be useful, though I have one major concern and several minor comments that I feel need to be addressed before publication.

Major comment:

The Angstrom exponent, used for deriving the aerosol index, is notoriously bad over land for MODIS Dark Target land retrieval (not sure about Deep Blue retrievals, and I'm unsure about which retrieval products were used here, though I believe that the 3 land regions used in this paper will mostly be Dark Target rather than Deep Blue). In the Levy et al. 2013 paper describing collection 6: "On a global basis, we and others have found little quantitative skill in MODIS-retrieved aerosol size parameters over land (e.g., Levy et al., 2010; Mielonen et al., 2011). We have decided to discontinue further attempts at validating Ångström Exponent (AE) and fine-AOD. A user can still choose to derive AE (from spectral AOD) or fine-AOD (from product of $\tau\ \eta$) and evaluate the results themselves." In Levy et al. 2010, they show that the Rˆ2 of the Angstrom exponent over land is only 0.3 with AERONET.

Levy et al. 2013: https://www.atmos-meas-tech.net/6/2989/2013/amt-6-2989-2013.pdf
Levy et al. 2010: https://www.atmos-chem-phys.net/10/10399/2010/acp-10-10399-2010.pdf

Related, for research on estimating surface-level PM2.5 from AOD, it would be useful to use the Angstrom exponent or fine-mode AOD products to help filter coarse PM from the PM2.5 estimate. However, due to the poor ability for MODIS to retrieve these products over land, the AOD-PM2.5 community uses just AOD as using the Angstrom exponent or fine-mode products adds noise to evaluations. It is unclear why the aerosol-cloud interactions community should have more confidence using the MODIS Angstrom exponent over land.

It seems that the aerosol index over land used here may be very poor (due to the Angstrom exponent uncertainty_, and this could be contributing to counterintuitive results over land. There is no mention of this in the paper. I feel that the authors should evaluate their aerosol index data in their 3 regions against AERONET to show if their aerosol index product has any skill in their land regions, and discussion needs to be added.

Minor comments

L29: "Earth" should be "Earth's"

L31-33: It is more precise to say that ACIs are the largest forcing uncertainty to present climate change. I'm nearly certain that the uncertainty in what humans will do (e.g. in terms of how much CO2 we will emit and other changes) is the biggest uncertainty in the anthropogenic contribution to *future* climate change.

L50-51: Kohler theory only describes the relationship between a particle and its critical supersaturation for activation. Cloud droplet number at the cloud base depends on the number distribution of critical supersaturations and the updraft velocity. The second part of the sentence discusses these effects, but I feel that this sentence can be rewritten to be more precise.

L64: "Loosely related". Is it possible to be more quantitative here (e.g. using numbers from the Stier reference)?

L68: Same, can you be quantitative?

L125: Which MODIS AOD retrieval algorithms are you using (dark target, deep blue, MAIAC)?

L268: "Become*s*"

L269: Either "*the* collision-coalescence process" or "collision-coalescence process*es*".

Figure 7 and associated discussion: It is unsurprising that the slope of CER-AI is stronger when including all data vs. when binning to >14um and <14um. If the slope

is positive, more observations will move from the <14um to >14um with increasing AI. With increasing AI, a CER of 13.9 um (which would be at the large end of the <14 um bin) would shift to 14.1 um (which would be on the low end of the >14 um bin), so increasing AI removes some of the higher <14um cases and creates new lower >14 um cases. . . the all prevents the CER for the two size bins from change much with AI. Hence, the slopes for the binned sizes are buffered from changes with AI. A similar phenomena will occur of a negative slope. Thus, the discussion in lines 274-278 seems unnecessary.